# UNLEASHING FLOW POLICIES WITH DISTRIBUTIONAL CRITICS

## ABSTRACT

Flow-based policies have recently emerged as a powerful tool in offline and offline-to-online reinforcement learning, capable of modeling the complex, multimodal behaviors found in pre-collected datasets. However, the full potential of these expressive actors is often bottlenecked by their critics, which typically learn a single, scalar estimate of the expected return. To address this limitation, we introduce the Distributional Flow Critic (DFC), a novel critic architecture that learns the complete state-action return distribution. Instead of regressing to a single value, DFC employs flow matching to model the distribution of return as a continuous, flexible transformation from a simple base distribution to the complex target distribution of returns. By doing so, DFC provides the expressive flow-based policy with a rich, distributional Bellman target, which offers a more stable and informative learning signal. Extensive experiments across D4RL and OG-Bench benchmarks demonstrate that our approach achieves strong performance, especially on tasks requiring multimodal action distributions, and excels in both offline and offline-to-online fine-tuning compared to existing methods.

## 1 INTRODUCTION

In modern reinforcement learning, particularly in offline and offline-to-online settings, a central challenge is learning effective policies from complex, pre-collected datasets (Fujimoto & Gu, 2021; Tarasov et al., 2023b; Park et al., 2025b). To this end, flow-based policies, trained with generative techniques like flow matching, represent a significant advance (Lipman et al., 2023; Zhang et al., 2025). Unlike traditional parameterizations, such as a unimodal Gaussian, flow-based methods can capture the rich, often multimodal action distributions inherent in diverse expert data. The success of such agents is measured not only by their static offline performance but also by their capacity for safe and efficient online fine-tuning.

Despite the success of these expressive policies, a critical limitation persists in their underlying learning mechanism. Current state-of-the-art flow-based methods pair a highly expressive actor (trained with flow matching) with a comparatively simple critic that only estimates the expected value of future returns, $Q(s, a)$ (Dabney et al., 2018b). This creates an "information bottleneck": the critic provides a single scalar value as a learning signal, which is a low-dimensional summary of a potentially complex and stochastic future. This simplistic signal may be insufficient to effectively guide the training of a high-capacity, multimodal policy. We argue that for an expressive policy to reach its full potential, it requires an equally expressive critic. The critic should provide a richer, more informative signal that captures not just the expected outcome, but the entire distribution of possible returns. This distributional perspective, pioneered by (Bellemare et al., 2017; Dabney et al., 2018a; Morimura et al., 2010), is known to improve learning stability and performance.

To bridge this crucial gap, we introduce the **Distributional Flow Critic (DFC)**, a novel architecture designed to learn the entire return distribution, $Z(s, a)$, using flow matching. However, a naive application of flow matching to established distributional losses faces a significant hurdle: estimating the loss requires sampling from the learned distribution $Z_\phi(s, a)$, which necessitates backpropagating through the entire trajectory of an ODE solver. This process is known to be computationally expensive and notoriously unstable, a challenge also observed in flow-based policy learning (Park et al., 2025c). To circumvent this, we propose an elegant **distillation-based architecture** comprising two critic networks. A multi-step *target flow critic* captures the complex target distribution,

while a single-step critic efficiently distills this distributional knowledge, ensuring stable and effective training. To ground our approach in a strong and relevant context, we integrate DFC into the Flow Q-learning framework (Park et al., 2025c), a state-of-the-art method known for its robust performance in offline and offline-to-online tasks. This allows our distributional critic to provide a rich, nuanced learning signal to an already powerful flow-based actor. Although our experiments focus on this specific integration, DFC is designed as a modular component. This suggests its potential as a drop-in replacement for standard scalar critics in other off-policy actor-critic algorithms, particularly those that also aim to capture complex, multimodal policies.

By creating a symbiotic architecture where both the actor and the critic leverage the power of flow matching, our method effectively addresses the dual challenges of behavioral complexity and value uncertainty. To validate our central claim, *"A distributional critic unlocks superior performance for flow-based policies."*, we benchmark DFC against a suite of state-of-the-art flow-policy methods. We conduct extensive evaluations on both pure offline and challenging offline-to-online benchmarks. Our results show that our method not only excels in the static offline phase but also demonstrates superior adaptability and sample efficiency during online fine-tuning. This confirms that equipping expressive policies with an equally expressive distributional critic is a critical step forward for both offline pre-training and subsequent online adaptation.

Our main contributions are:

**A Distributional Flow-Based Critic.** We introduce a flow-matching model for the critic that moves beyond traditional scalar Q-value estimation. This model learns to generate the entire distribution of future returns, providing a richer, more stable, and more informative learning signal for policy optimization, which is especially beneficial in highly stochastic environments. We conduct an ablation study to demonstrate the necessity of our proposed design choices.

**State-of-the-Art Empirical Performance.** Through extensive experiments on a wide range of challenging benchmarks, we demonstrate that our method significantly outperforms existing approaches, particularly on tasks that require multimodal action distributions. We also show that our framework is highly effective for offline-to-online fine-tuning. We empirically show that our method consistently outperforms existing state-of-the-art approaches, registering a comprehensive performance gain of more than $10\%$.

## 2 BACKGROUND

### 2.1 DISTRIBUTIONAL REINFORCEMENT LEARNING

We model the agent-environment interaction as a Markov Decision Process (MDP) (Sutton & Barto, 2018), formally defined by the tuple $(\mathcal{S}, \mathcal{A}, P, r, \gamma)$. Here, $\mathcal{S}$ is the state space, $\mathcal{A}$ is the action space, $P : \mathcal{S} \times \mathcal{A} \to \Delta(\mathcal{S})$ is the transition probability function where $\Delta(\cdot)$ denotes the set of probability distributions over a space, $r : \mathcal{S} \times \mathcal{A} \to \mathbb{R}$ is the reward function, and $\gamma \in [0, 1)$ is the discount factor. An agent's behavior is described by a policy $\pi : \mathcal{S} \to \Delta(\mathcal{A})$, which maps states to a probability distribution over actions. The goal of the agent is to learn a policy that maximizes the expected discounted cumulative return: $J(\pi) = \mathbb{E}_{\tau \sim \pi} \left[ \sum_{t=0}^{\infty} \gamma^t r(s_t, a_t) \right]$, where the trajectory $\tau = (s_0, a_0, s_1, a_1, \dots)$ is generated by executing policy $\pi$ (i.e., $s_0 \sim p_0(s)$, $a_t \sim \pi(\cdot|s_t)$, and $s_{t+1} \sim P(\cdot|s_t, a_t)$). The action-value function $Q^\pi(s, a)$ is the expected return after taking action $a$ in state $s$ and subsequently following $\pi$. The Q-function satisfies the Bellman expectation equation: $Q^\pi(s, a) = \mathbb{E}_{s' \sim P(\cdot|s,a)} \left[ r(s, a) + \gamma \mathbb{E}_{a' \sim \pi(\cdot|s')}[Q^\pi(s', a')] \right]$.

Standard reinforcement learning focuses on estimating the expected value of the cumulative return, $Q^\pi(s, a)$. Distributional Reinforcement Learning (DRL) (Bellemare et al., 2017) extends this paradigm by learning the entire probability distribution of the return. The return, or sum of discounted future rewards, is treated as a random variable, denoted by $Z^\pi(s, a)$:

$$Z^\pi(s, a) = \sum_{t=0}^{\infty} \gamma^t r(S_t, A_t), \quad \text{where } S_0 = s, A_0 = a. \tag{1}$$

The core of DRL is the *distributional Bellman equation*, which states that the distribution of the return at the current state-action pair is related to the distribution of the return at the next state: $\mathcal{T}^\pi Z(s, a) \stackrel{D}{:=} r(s, a) + \gamma Z^\pi(S', A')$, where $\mathcal{T}^\pi$ denotes the *distributional Bellman operator*, $\stackrel{D}{=}$ denotes equality in distribution, $S' \sim \mathcal{P}(\cdot|s, a)$, and $A' \sim \pi(\cdot|S')$.

In practice, DRL algorithms learn a parameterized approximation $Z_\theta(s, a)$ of the true return distribution. For example, C51 (Bellemare et al., 2017) models the distribution with a discrete set of fixed supports (atoms) and learns the corresponding probabilities. Quantile Regression DQN (QR-DQN) (Dabney et al., 2018b) takes a different approach by directly learning the quantile function of the return distribution.

A natural metric for comparing return distributions is the **Wasserstein distance** (Villani, 2003). Minimizing the distributional Bellman error with respect to the 1-Wasserstein distance, $W_1(\mathcal{T}^\pi Z_\theta(s, a), Z_\theta(s, a))$, is a primary objective. For one-dimensional distributions $\mu_X$ and $\mu_Y$ with cumulative distribution functions (CDFs) $F_X$ and $F_Y$, this distance has a closed-form solution: $W_1(\mu_X, \mu_Y) = \int_0^1 |F_X^{-1}(\tau) - F_Y^{-1}(\tau)| d\tau$. Given $N$ i.i.d. samples $\{x_i\}_{i=1}^N$ from $\mu_X$ and $\{y_i\}_{i=1}^N$ from $\mu_Y$, the distance can be approximated by sorting the samples (denoted by $\tilde{x}_i, \tilde{y}_i$): $\hat{W}_1(\mu_X, \mu_Y) \approx \frac{1}{N} \sum_{i=1}^N |\tilde{x}_i - \tilde{y}_i|$. However, this sample-based approximation is biased. QR-DQN elegantly circumvents this issue. Instead of directly minimizing an approximated Wasserstein distance, it minimizes the **quantile regression loss** (Koenker & Bassett, 1978). This objective implicitly minimizes the Wasserstein distance between the empirical distributions of samples, providing a more stable and effective method for learning the return distribution.

## 2.2 FLOW MATCHING AND FLOW Q-LEARNING

Flow Matching is a technique for training expressive generative models by learning a vector field $v_\theta$ that defines a mapping from a simple noise distribution to a complex target distribution by solving an ordinary differential equation (ODE) (Lipman et al., 2023). This framework can be used to create highly expressive *flow policies*, $\pi_\theta(a|s)$, capable of representing complex, multimodal action distributions, which is a significant advantage over simpler parametric forms like Gaussians. However, directly training these policies with reinforcement learning objectives is challenging, as it requires backpropagating gradients through the ODE solver, a process that is computationally expensive and notoriously unstable.

Flow Q-Learning (FQL) (Park et al., 2025c) is an offline and offline-to-online reinforcement learning method that elegantly solves this problem by decoupling the policy's training from direct value maximization. The core idea is to first train an iterative flow policy using only a behavioral cloning (BC) objective on the offline dataset. This BC flow policy, denoted $\mu_\theta(s, \epsilon)$ where $\epsilon$ is the initial noise, learns to capture the complex action distribution of the dataset. While $\mu_\theta$ is deterministic function, the random sampling of $\epsilon$ allows it to function as a stochastic policy, formally expressed as $\pi_\theta(a \mid s)$. In this paper, we slightly abuse notation for simplicity and refer to both $\mu_\theta$ and $\pi_\theta$ as the "policy".

Instead of directly optimizing this iterative policy with RL, FQL introduces a separate, expressive one-step policy, $\pi_\omega$, which is trained to maximize Q-values while being regularized via distillation from the BC flow policy. This distillation loss prevents the one-step policy from deviating too far from the learned data distribution. The distillation loss is defined as the mean squared error between the two policies' outputs for a given state and noise vector:

$$\mathcal{L}_{\text{Distill}}(\omega) = \mathbb{E}_{s \sim \mathcal{D}, \epsilon \sim \mathcal{N}(0, I)} \left[ \|\pi_\omega(s, \epsilon) - \mu_\theta(s, \epsilon)\|^2 \right] \tag{2}$$

The complete objective for the one-step policy $\pi_\omega$ combines the Q-maximization term with this distillation loss, which acts as a behavioral cloning regularizer:

$$\mathcal{L}_\pi(\omega) = \mathbb{E}_{s \sim \mathcal{D}, a^\pi \sim \pi_\omega}[-Q_\phi(s, a^\pi)] + \alpha \mathcal{L}_{\text{Distill}}(\omega) \tag{3}$$

This approach allows FQL to leverage the expressivity of the flow model while using a stable, one-step update for the RL objective, thereby avoiding unstable backpropagation through the ODE solver.

## 3 METHODOLOGY

Our core contribution is a novel distributional critic based on flow matching, designed to enhance the expressive power of any actor-critic method that employs a flow-based policy. In principle, our critic can be integrated into various such frameworks. To demonstrate its effectiveness, we build upon the state-of-the-art Flow Q-Learning (FQL) framework, chosen for its proven stability and performance.

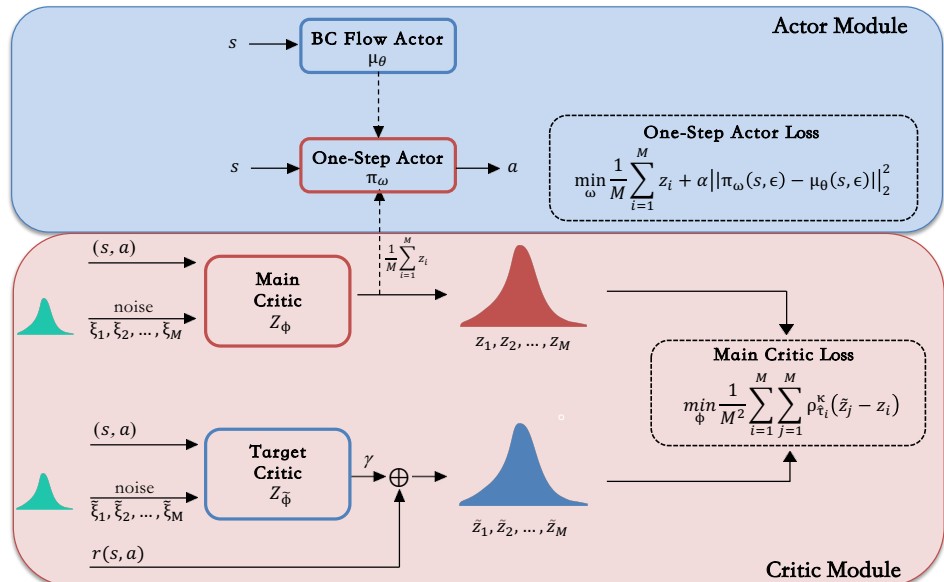

Figure 1: **A schematic of the DFC architecture.** The illustration delineates the decoupled training paradigm, wherein the target distributional flow critic, $Z_{\tilde{\phi}}$, is updated to form a set of target return samples, which in turn supervises the main distributional critic, $Z_\phi$. The synergistic output of both critics is then utilized to optimize the dual-policy actor.

## 3.1 ACTOR ARCHITECTURE

For our actor, we directly adopt the dual-policy architecture from Flow Q-Learning (FQL). As previously described, this framework uses a one-step policy, $\pi_\omega$, which is updated to maximize Q-values while being regularized via distillation from an expressive, behaviorally-cloned flow policy, $\mu_\theta$. The objective function is similar to Equation 3, and will elaborate how to estimate it after we define the critic later. While we retain this actor structure, our core contribution lies in the novel formulation of the critic function $Q_\phi(s, a^\pi)$, which is designed to match the actor's expressive power and is detailed next.

## 3.2 CRITIC AS A CONDITIONAL DISTRIBUTIONAL FLOW MODEL

A critical limitation in actor-critic methods arises when an expressive actor is paired with a simple critic. A standard critic that predicts only the mean of the return cannot capture the potentially complex value distributions—such as multimodal or skewed returns—that a powerful actor might induce. This mismatch can lead to an impoverished learning signal and unstable training.

To address this, we propose a distributional critic that learns the entire distribution of returns. A naive approach would be to model the critic $Z_\phi(s, a)$ as a single conditional flow model trained to minimize the Wasserstein distance to the target distribution, e.g., $W_1(\mathcal{T}^\pi Z_\phi(s, a), Z_\phi(s, a))$. However, this approach is infeasible for two key reasons:
**Unstable Critic Training.** Generating samples from $Z_\phi$ to compute the loss would require solving an ODE. Training would thus necessitate backpropagation through the ODE solver's steps, which is notoriously unstable.
**Unstable Actor Training.** The actor update requires the gradient $\nabla_a Q(s, a)$. If the Q-value is the mean of the flow-generated distribution $Z_\phi$, computing this gradient would also require backpropagation through the ODE solver, which the very problem FQL was designed to avoid.

To circumvent this issue, we introduce a two-stage architecture comprising a **target distributional flow critic**, $Z_{\tilde{\phi}}$, and a **main distributional critic**, $Z_\phi$. This approach is analogous to knowledge

distillation: the powerful, multi-step flow model $Z_{\tilde{\phi}}$ learns the complex target distribution, and its knowledge is then distilled into the simpler, one-step critic $Z_\phi$, which can be updated efficiently.

**Learning the Target Distribution with Flow Matching.** The target critic $Z_{\tilde{\phi}}$ is a conditional flow model designed to approximate the target value distribution $\mathcal{T}^\pi Z(s,a)$. Its velocity field, $v_{\tilde{\phi}}$, is updated using the flow matching objective. Specifically, for a given transition $(s, a, r, s')$, we first construct a set of target return samples $\{\tilde{z}_j\}_{j=1}^{M}$ using the distributional Bellman equation:

$$\tilde{z}_j = r + \gamma z'_j, \quad \text{where} \quad z'_j = Z_{\tilde{\phi}}(s', \pi_\omega(s'), \tilde{\xi}_j). \tag{4}$$

Here, samples $z'_j$ are generated by solving the ODE defined by the target critic $Z_{\tilde{\phi}}$ itself (with frozen parameters from a previous iteration, akin to a standard target network). Then, the parameters $\tilde{\phi}$ are updated by minimizing the flow matching loss, which trains the model to transport samples from a base Gaussian distribution to this target distribution:

$$\mathcal{L}_{\text{Flow}}(\tilde{\phi}) = \mathbb{E}_{\tau, \tilde{\xi}, \tilde{z}} \left[ ||v_{\tilde{\phi}}(\tau, (1-\tau)\tilde{\xi} + \tau\tilde{z}|s, a) - (\tilde{z} - \tilde{\xi})||_2^2 \right], \tag{5}$$

where $\tau \sim \mathcal{U}(0,1)$, $\tilde{\xi} \sim \mathcal{N}(0, I_d)$, and $\tilde{z}$ is a sample from the target set constructed above.

**Distilling the Target Distribution via Quantile Regression.** After updating the target flow network, we train the main critic $Z_\phi$. The purpose of $Z_\phi$ is to learn a direct, one-step mapping to a distribution that matches the output of the target flow critic $Z_{\tilde{\phi}}$. This avoids the need for an ODE solver during the actor update. We achieve this distillation by minimizing the 1-Wasserstein distance between the two distributions, using the quantile Huber loss as a practical and robust proxy (Dabney et al., 2018b).

We draw a set of $M$ samples $\{\tilde{z}_j\}$ from the target critic $Z_{\tilde{\phi}}$ and another set of $M$ quantile values $\{z_i\}$ from the main critic $Z_\phi$. The main critic's parameters $\phi$ are then updated by minimizing the following loss:

$$\mathcal{L}_{\text{critic}}(\phi) = \frac{1}{M^2} \sum_{i=1}^{M} \sum_{j=1}^{M} \rho_{\hat{\tau}_i}^\kappa(\tilde{z}_j - z_i), \tag{6}$$

where $\rho_{\hat{\tau}}^\kappa$ is the quantile Huber loss function defined as:

$$\rho_{\hat{\tau}}^\kappa(u) = |\hat{\tau} - \mathbb{I}(u < 0)|\mathcal{L}_\kappa(u), \quad \text{with} \quad \mathcal{L}_\kappa(u) = \begin{cases} 0.5u^2 & \text{if } |u| \leq \kappa \\ \kappa(|u| - 0.5\kappa) & \text{otherwise.} \end{cases} \tag{7}$$

The quantile levels are set to $\hat{\tau}_i = (2i-1)/(2M)$ for $i = 1, \ldots, M$. This loss effectively trains $Z_\phi$ to produce a distribution that mirrors the one generated by the more complex target flow critic $Z_{\tilde{\phi}}$. The actor is then updated using the distilled critic $Z_\phi$, sidestepping the BPTT problem entirely.

Crucially, this distillation process via quantile regression is not merely a heuristic approximation. It can be theoretically formalized as a projection operator that preserves the fundamental stability properties of the Bellman update. To rigorously justify the convergence of our two-stage critic, we establish that the combined update operator maintains the contraction property essential for fixed-point iteration.

To rigorously analyze the convergence properties of our critic, we first establish the metric space over distributional value functions. Let $\mathcal{Z}$ denote the space of value distributions mapping state-action pairs to probability measures over $\mathbb{R}$. We utilize the *maximal 1-Wasserstein metric* Bellemare et al. (2017), denoted as $\bar{d}_\infty$, which is defined as the supremum of the 1-Wasserstein distance over the entire state-action space:

$$\bar{d}_\infty(Z_1, Z_2) := \sup_{s \in \mathcal{S}, a \in \mathcal{A}} W_1(Z_1(s,a), Z_2(s,a)),$$

where $W_1(\cdot, \cdot)$ is the standard 1-Wasserstein distance between two probability distributions. An operator $\mathcal{T} : \mathcal{Z} \to \mathcal{Z}$ is defined as a $\gamma$-*contraction* with respect to $\bar{d}_\infty$ if there exists a constant $\gamma \in [0, 1)$ such that for any two distributional value functions $Z_1, Z_2 \in \mathcal{Z}$: $\bar{d}_\infty(\mathcal{T}Z_1, \mathcal{T}Z_2) \leq \gamma \bar{d}_\infty(Z_1, Z_2)$.

Standard distributional RL theory (Bellemare et al., 2017) established that the distributional Bellman operator $\mathcal{T}^\pi$ is a $\gamma$-contraction under this metric ($\gamma$ is the discount factor in MDP). Building on this, we show that our specific distillation process preserves this property.

**Proposition 1.** *Let $\mathcal{T}^\pi$ be the distributional Bellman operator and $\Pi_{W_1}$ be the projection operator onto the space of quantile distributions via 1-Wasserstein minimization. The combined distillation operator, defined as $\hat{\mathcal{T}} := \Pi_{W_1}\mathcal{T}^\pi$, constitutes a $\gamma$-contraction mapping with respect to the maximal 1-Wasserstein metric $\bar{d}_\infty$. That is, for any two return distributions $Z_1, Z_2$:*

$$\bar{d}_\infty(\hat{\mathcal{T}}Z_1, \hat{\mathcal{T}}Z_2) \leq \gamma\bar{d}_\infty(Z_1, Z_2), \tag{8}$$

Following the same analytical techniques in (Bellemare et al., 2017; Dabney et al., 2018b), we defer the proof to Appendix D.

### 3.3 ACTOR UPDATE

With our trained main critic $Z_\phi$, we can now update the actor policy $\pi_\omega$. Following common practice in distributional RL (Barth-Maron et al., 2018b), we estimate the Q-value as the mean of the return distribution. The actor $\pi_\omega$ is then trained to produce actions that maximize this estimated Q-value. The final objective for the actor combines the Q-maximization term with the FQL distillation regularizer:

$$\mathbb{E}[-\frac{1}{M}\sum_{i=1}^{M}[Z_\phi(s, \mu_\omega(s, \epsilon), \xi_i)] + \alpha\|\mu_\omega(s, \epsilon) - \mu_\theta(s, \epsilon)\|_2^2].$$

Because $Z_\phi$ is a simple feed-forward network, computing the policy gradient is stable and efficient, entirely sidestepping the BPTT problem.

All the steps of our algorithm are described in Algorithm 1. The synergistic integration of our distributional critic and dual-policy actor forms a holistic framework with significant advantages. The flow-based critic architecture allows us to model the entire value distribution with unprecedented fidelity, capturing nuances such as multimodality and skewness that are invisible to traditional, expectation-based critics. This rich distributional information provides a more robust and informative gradient signal to the actor, moving beyond a simple scalar reward signal. The design not only enhances training stability but also allows each component to leverage its respective strengths: the expressiveness of flow models for target generation and the efficiency of quantile regression for distributional matching. The result is a robust framework for accurately approximating complex value distributions in reinforcement learning.

## 4 RELATED WORKS

### 4.1 DIFFUSION AND FLOW POLICY RL

Recent advancements in iterative generative modeling, particularly denoising diffusion (Sohl-Dickstein et al., 2015; Ho et al., 2020) and flow matching (Lipman et al., 2023; Esser et al., 2024), have spurred significant innovation within reinforcement learning (RL) due to their capacity to model complex, high-dimensional data distributions.

Previous works can be broadly categorized based on how they leverage these generative models. One line of research applies them to high-level decision-making tasks such as planning and hierarchical learning (Janner et al., 2022; Ajay et al., 2023; Zheng et al., 2023; Liang et al., 2023; Suh et al., 2023; Venkatraman et al., 2024; Chen et al., 2024a). Another prominent application is to use generative models create synthetic environments or supplement training data to improve policy robustness and generalization (Lu et al., 2023b; Ding et al., 2024; Jackson et al., 2024; Alonso et al., 2024). A third area involves the use of generative models to improve strategy implementation (Mazoure et al., 2020; Ren et al., 2024) or to model policies (Park et al., 2025c).

Several strategies have been proposed to train flow and diffusion policies for offline reinforcement learning. These can be broadly categorized into three main paradigms: 1) **Value-weighted regression**, which prioritizes high-advantage actions but can be limited by the expressivity of policies trained on static datasets (Lu et al., 2023a; Kang et al., 2023; Zhang et al., 2025); 2) **Reparameterization-based policy gradients**, which optimize the value function by backpropagating through the policy but often suffer from instability and high variance (Wang et al., 2023; He et al., 2023; Ding & Jin, 2024b); and 3) **Rejection sampling**, which enhances stability by filtering actions but incurs substantial computational overhead (Chen et al., 2023; Hansen-Estruch

---

**Algorithm 1** Distributional Flow Critic (DFC)

---

**for** the number of environment steps **do**
    Sample batch $\{(s, a, r, s')\}$
    ▷ Train vector field $v_{\tilde{\phi}}$ in distributional flow critic $Z_{\tilde{\phi}}$
    Sample noise $\tilde{\xi}_1, \tilde{\xi}_2, \ldots, \tilde{\xi}_M \sim \mathcal{N}(0, I_d)$
    $\tau \sim \mathcal{U}(0, 1)$
    **for** $j = 1, \ldots, M$ **do**
        $\tilde{z}_j \leftarrow r + \gamma Z_{\tilde{\phi}}(s', \pi_\omega(s'), \tilde{\xi}_j)$         ▷ Bellman Update
        $\tilde{z}_j^\tau \leftarrow (1 - \tau)\tilde{\xi}_j + \tau\tilde{z}_j$
    Update $\tilde{\phi}$ to minimize $\mathbb{E}\left[\frac{1}{M}\sum_{j=1}^M \|v_{\tilde{\phi}}(\tau, s, \tilde{z}_j^\tau) - (\tilde{z}_j - \tilde{\xi}_j)\|_2^2\right]$
    ▷ Train distributional critic $Z_\phi$
    Sample noise $\xi_0, \xi_1, \ldots, \xi_{M-1} \sim \mathcal{N}(0, I_d)$,
    $z_j \leftarrow Z_\phi(s, a, \xi_j)$ for $j = 1, \ldots, M$
    Update $\phi$ to minimize $\mathbb{E}\left[\frac{1}{M^2}\sum_{i=1}^M\sum_{j=1}^M \rho_{\hat{\tau}_i}^\zeta(\tilde{z}_j - z_i)\right]$
    ▷ Train vector field $v_\theta$ in flow policy $\pi_\theta$
    Sample noise $x^0 \sim \mathcal{N}(0, I_d)$
    $x^1 \leftarrow a$
    Sample $t \sim \mathcal{U}([0, 1])$
    $x^t \leftarrow (1 - t)x^0 + tx^1$
    Update $\theta$ to minimize $\mathbb{E}[\|v_\theta(t, s, x^t) - (x^1 - x^0)\|_2^2]$
    ▷ Train one-step policy $\pi_\omega$
    Sample noise $\epsilon \sim \mathcal{N}(0, I_d)$
    Update $\omega$ to minimize $\mathbb{E}[-\frac{1}{M}\sum_{i=1}^M [Z_\phi(s, \mu_\omega(s, \epsilon), \xi_i)] + \alpha\|\mu_\omega(s, \epsilon) - \mu_\theta(s, \epsilon)\|_2^2]$
**return** One-step policy $\pi_\omega$

---

et al., 2023; He et al., 2024). Beyond these, other techniques include direct action gradients (Yang et al., 2023; Psenka et al., 2024; Li et al., 2024), bi-level MDP formulations (Ren et al., 2024), and integrations with implicit Q-learning (Chen et al., 2024b;c).

In this work, we employ a reparameterized policy gradient approach. Distinct from prior methods such as Diffusion-QL (Wang et al., 2023), DiffCPS (He et al., 2023), Consistency-AC (Ding & Jin, 2024b), SRDP (Ada et al., 2024), and EQL (Zhang et al., 2024), our method circumvents backpropagation through the generative model by instead training a one-step policy alongside a distributional critic. This strategy notably improves training stability and computational efficiency. Empirically, we demonstrate that our approach achieves a significant performance leap over existing baselines.

## 4.2 DISTRIBUTIONAL RL

Distributional Reinforcement Learning (DRL) marked a paradigm shift from modeling expected returns to capturing the full distribution of stochastic outcomes (Bellemare et al., 2017; Dabney et al., 2018b; Qu et al., 2019). Early value-based methods for discrete action spaces, such as C51 (Bellemare et al., 2017), parameterized the return distribution with discrete atoms. This was advanced by QR-DQN (Dabney et al., 2018b) and IQN (Dabney et al., 2018a), which respectively used discrete quantiles and learned a full continuous quantile function for enhanced flexibility. The extension of DRL to continuous control yielded policy-gradient methods like D4PG (Barth-Maron et al., 2018a), which demonstrated strong performance but inherited the representational limitations of a discrete categorical critic. To overcome this constraint, which makes it difficult to capture complex return distributions, SDPG (Singh et al., 2022) introduced a sample-based distributional policy gradient method that models the return distribution via reparameterization, thus avoiding the constraints of discrete representations.

Building upon this trajectory, our work further leverages flow matching as the core generative mechanism for modeling distributions in RL. Unlike previous sample-based methods that often rely on adversarial training, our approach benefits from the unique advantages inherent to flow matching. Firstly, flow matching offers superior expressiveness by directly learning a continuous-time vector

field that transforms a simple noise distribution into any arbitrarily complex target distribution. By integrating this powerful generative tool into both the critic and actor, our method provides a robust and highly flexible framework for distributional reinforcement learning.

# 5 EXPERIMENT

## 5.1 BENCHMARKS

Our empirical evaluation is conducted on two comprehensive benchmark suites: the widely-adopted D4RL benchmark (Fu et al., 2020) and the recently proposed OGBench (Park et al., 2025a), which together provide a diverse set of challenges spanning locomotion, manipulation, and both state-based and pixel-based observations.

**D4RL.** From the D4RL benchmark, we select a suite of particularly challenging tasks to rigorously test our algorithm's capabilities. Specifically, we evaluate on six distinct `antmaze` navigation tasks and twelve `adroit` manipulation tasks. For `adroit` tasks, performance is reported using the standard normalized return score (Fu et al., 2020), calculated as: Normalized Return $= 100 \times \frac{\text{score}-\text{random\_score}}{\text{expert\_score}-\text{random\_score}}$. For `antmaze` tasks, we report the success rate, which measures the percentage of episodes where the agent successfully completes the designated goal.

**OGBench.** To assess the generalization capabilities of our method, we also employ the OGBench suite. Our evaluation includes 50 state-based tasks, comprising five locomotion and five manipulation environments, each with five distinct dataset compositions. Additionally, we evaluate on five challenging visual manipulation tasks from OGBench. For all OGBench tasks, the primary evaluation metric is the success rate.

## 5.2 METHODS

To furnish a rigorous and comprehensive comparative analysis, our experiments are conducted across both offline and online reinforcement learning settings. The selection of baseline algorithms encompasses a diverse spectrum of policy classes and training paradigms, enabling a thorough evaluation of our proposed method.

**Offline RL Baselines.** In the offline learning context, our empirical comparison is organized around three principal categories of algorithms. For Gaussian policies, we select Implicit Q-Learning (IQL) (Kostrikov et al., 2022) and Regularized Behavior-Cloning with Active-Critic (ReBRAC) (Tarasov et al., 2023a) as strong, standard baselines that employ simple, unimodal policies. To benchmark against diffusion policies, we include IDQL (Hansen-Estruch et al., 2023), which leverages rejection sampling for policy improvement, and Consistency-AC (CAC) (Ding & Jin, 2024a), which implements policy distillation via backpropagation through a consistency model. Finally, within the domain of flow policies, we benchmark against two distinct strategies: Implicit Flow Q-Learning (IFQL) (Park et al., 2025c), a flow-based counterpart to IDQL, and Flow Q-Learning (FQL) (Park et al., 2025c), which represents the state-of-the-art in this class by training an efficient one-step policy via distillation.

**Offline-to-Online RL Baselines.** For the online fine-tuning phase, our evaluation focuses on algorithms capable of effectively adapting from an offline-pretrained policy. We compare our method against a curated subset of the aforementioned baselines amenable to online interaction, namely IQL, ReBRAC, IFQL, and FQL. This experimental design allows for a direct assessment of both the asymptotic performance and sample efficiency of our approach during online adaptation.

A detailed specification of the hyperparameter configurations for all algorithms and experimental setups is provided in Section A.1 to ensure full reproducibility.

## 5.3 RESULTS

Table 1 presents a comprehensive summary of our aggregated benchmark results across a total of 73 state- and pixel-based offline RL tasks, spanning both robotic locomotion and manipulation. The results indicate that DFC exhibits superior performance over the vast majority of established methods, including those predicated on Gaussian, diffusion, and flow-based policies.

Figure 2 further illustrates the performance gains achieved by our method in both offline and offline-to-online settings, demonstrating consistent improvements across most tasks. Our approach substantially outperforms previous methods, especially on some tasks of medium difficulty, while also elevating performance on the majority of other tasks. Particularly noteworthy is DFC's performance on some of the most challenging tasks within the D4RL benchmark, achieving scores of $91\%$ and $95\%$ on `antmaze-large-play` and `antmaze-large-diverse`, respectively.

To evaluate DFC's efficacy for fine-tuning, we transition from the offline pre-training phase by incorporating newly collected online transitions into the replay buffer. Following the protocol of FQL, online fine-tuning commences after one million offline gradient steps, continuing to optimize all network components with the same objectives. The evaluation is conducted on the same suite of 68 state-based RL tasks used in our offline experiments.

To underscore the criticality of our proposed architecture, we conduct an ablation study with two simplified variants. The first, a **Flow-only Critic (FC)** (Removes the distillation step and attempts to update the actor directly using the flow-based critic), suffers from significant training instability and high variance across different seeds, rarely surpassing our full model's performance. The reason is that this approach necessitates backpropagating gradients through the ODE solver's trajectory, a process known to be computationally expensive and notoriously unstable. The second, a **Distributional-only Critic (DC)** (Relies on standard quantile regression without the generative flow-based target), demonstrates stable training but yields suboptimal performance, offering only marginal improvements over the FQL baseline in limited scenarios. The reason is that it lacks the generative expressivity required to capture the complex, multimodal return distributions. These results, detailed in Table 5 in Appendix B , highlight that while the distributional component ensures stability, it is insufficient on its own, and the combination in our two-stage architecture is essential for achieving state-of-the-art performance.

In contrast to these ablated versions, our full **DFC** model consistently delivered the best performance. It excelled across a diverse range of challenging domains, including robotic locomotion, manipulation, offline reinforcement learning, and offline-to-online fine-tuning, in both state- and pixel-based settings. Notably, DFC achieves these results without the computational burden of backpropagation through time, highlighting its efficiency and effectiveness. More ablation results are listed in Appendix B.

# 6 CONCLUSION

In this work, we introduced the Distributional Flow Critic (DFC), a novel critic architecture that synergistically integrates the predictive power of flow matching with the stability of distributional reinforcement learning. Our empirical evaluation demonstrates that this fusion overcomes the training instability inherent in a pure flow-based critic while surpassing the performance limitations of a purely distributional one. The consistent state-of-the-art results across a wide array of challenging benchmarks underscore the efficacy and robustness of our approach. By successfully unifying these two powerful paradigms, DFC establishes a new direction for designing high-performance and

Table 1: **Offline RL results.** DFC achieves the best or near-best performance on most of the 73 diverse, challenging benchmark tasks. The performances are averaged over 8 seeds (4 seeds for pixel-based tasks). The best scores are highlighted in **bold** and the suboptimal scores are labeled as underlined. Tarasov et al. (2023b); Hansen-Estruch et al. (2023); Chen et al. (2024b) contribute to the cells without the "$\pm$" sign. See Table 4 for the full results in Appendix B.

| Task Category | IQL | ReBRAC | IDQL | CAC | IFQL | FQL | DFC |
|---|---|---|---|---|---|---|---|
| OGBench antmaze-large-singletask (5 tasks) | $53_{\pm3}$ | $\underline{81}_{\pm5}$ | $21_{\pm5}$ | $33_{\pm4}$ | $28_{\pm5}$ | $79_{\pm3}$ | $\mathbf{88}_{\pm2}$ |
| OGBench antmaze-giant-singletask (5 tasks) | $4_{\pm1}$ | $\mathbf{26}_{\pm8}$ | $0_{\pm0}$ | $0_{\pm0}$ | $3_{\pm2}$ | $9_{\pm6}$ | $\underline{19}_{\pm14}$ |
| OGBench humanoidmaze-medium-singletask (**5** tasks) | $33_{\pm2}$ | $22_{\pm8}$ | $1_{\pm0}$ | $53_{\pm8}$ | $\underline{60}_{\pm14}$ | $58_{\pm5}$ | $\mathbf{67}_{\pm14}$ |
| OGBench humanoidmaze-large-singletask (5 tasks) | $2_{\pm1}$ | $2_{\pm1}$ | $1_{\pm0}$ | $0_{\pm0}$ | $\underline{11}_{\pm2}$ | $4_{\pm2}$ | $\mathbf{19}_{\pm3}$ |
| OGBench antsoccer-arena-singletask (5 tasks) | $8_{\pm2}$ | $0_{\pm0}$ | $12_{\pm4}$ | $2_{\pm4}$ | $33_{\pm6}$ | $\underline{60}_{\pm2}$ | $\mathbf{73}_{\pm4}$ |
| OGBench cube-single-singletask (5 tasks) | $83_{\pm3}$ | $91_{\pm2}$ | $95_{\pm2}$ | $85_{\pm9}$ | $79_{\pm2}$ | $\underline{96}_{\pm1}$ | $\mathbf{100}_{\pm0}$ |
| OGBench cube-double-singletask (5 tasks) | $7_{\pm1}$ | $12_{\pm1}$ | $15_{\pm6}$ | $6_{\pm2}$ | $14_{\pm3}$ | $\underline{29}_{\pm2}$ | $\mathbf{41}_{\pm4}$ |
| OGBench scene-singletask (5 tasks) | $28_{\pm1}$ | $41_{\pm3}$ | $46_{\pm3}$ | $40_{\pm7}$ | $30_{\pm3}$ | $\underline{56}_{\pm2}$ | $\mathbf{63}_{\pm2}$ |
| OGBench puzzle-3x3-singletask (5 tasks) | $9_{\pm1}$ | $21_{\pm1}$ | $10_{\pm2}$ | $19_{\pm0}$ | $19_{\pm1}$ | $\underline{30}_{\pm1}$ | $\mathbf{40}_{\pm2}$ |
| OGBench puzzle-4x4-singletask (5 tasks) | $7_{\pm1}$ | $14_{\pm1}$ | $\underline{29}_{\pm3}$ | $15_{\pm3}$ | $25_{\pm5}$ | $17_{\pm2}$ | $\mathbf{35}_{\pm4}$ |
| D4RL antmaze (6 tasks) | $57$ | $78$ | $79$ | $30_{\pm3}$ | $65_{\pm7}$ | $\underline{84}_{\pm3}$ | $\mathbf{92}_{\pm2}$ |
| D4RL adroit (12 tasks) | $53$ | $\underline{59}$ | $52_{\pm1}$ | $43_{\pm2}$ | $52_{\pm1}$ | $52_{\pm1}$ | $\mathbf{63}_{\pm3}$ |
| Visual manipulation (5 tasks) | $42_{\pm4}$ | $60_{\pm2}$ | - | - | $50_{\pm5}$ | $\underline{65}_{\pm2}$ | $\mathbf{68}_{\pm6}$ |

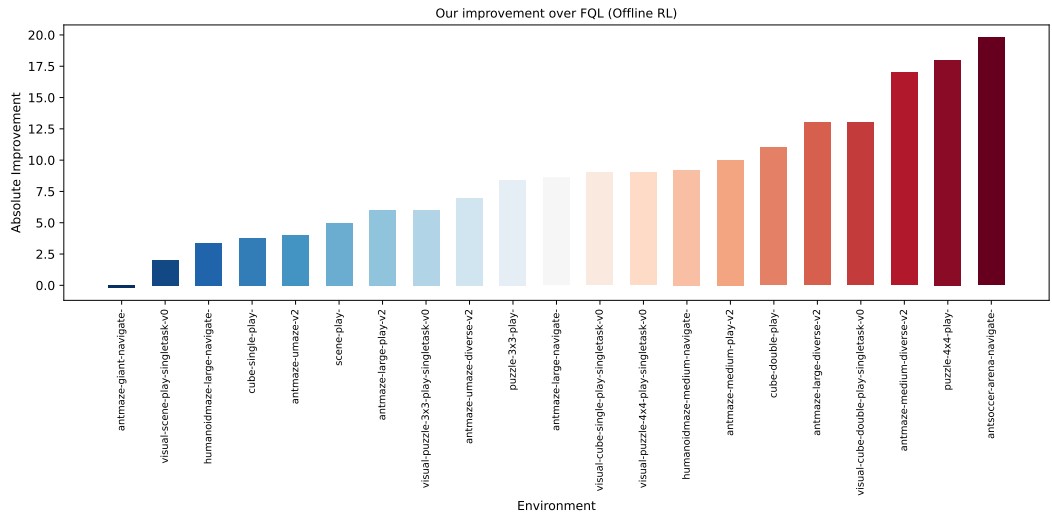

(a) Offline Phase Results Difference

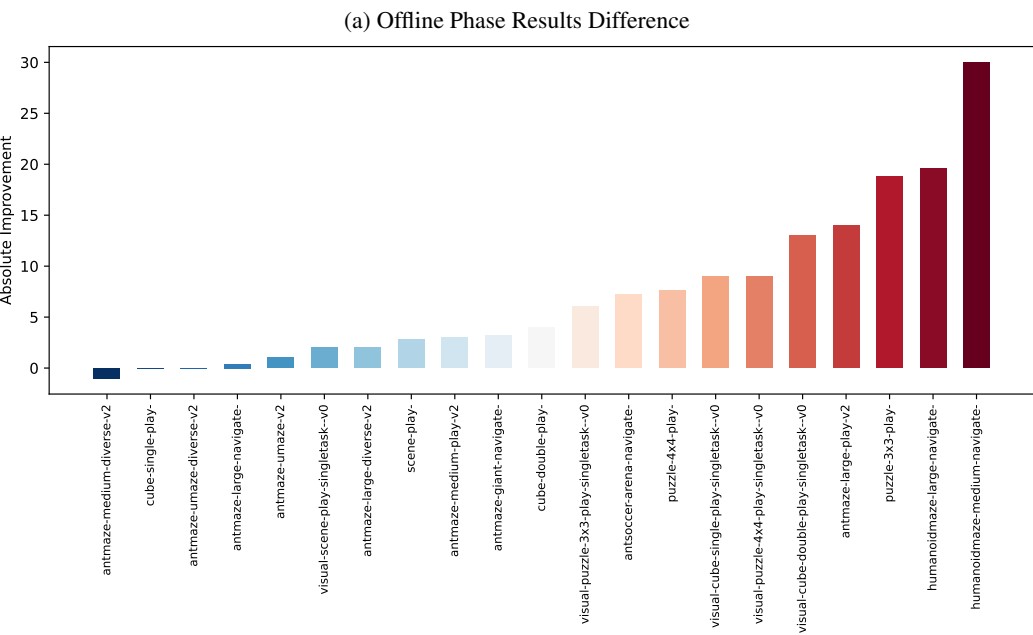

(b) Offline+Online Phase Results Difference

Figure 2: **Performance Comparison between DFC and FQL.** Conducted on tasks evaluated by success rates over 8 seeds.

stable critics in reinforcement learning. Future work could explore the application of DFC to more complex, partially observable environments or its integration with hierarchical learning frameworks.

## ETHICS STATEMENT

This work adheres to the ICLR Code of Ethics. All experiments were conducted in simulated environments, involving no direct interaction with human subjects or animals. The datasets used in this research are standard, publicly available benchmarks, and their use is consistent with their original licensing and intended purpose. We have made efforts to ensure that our research promotes scientific excellence and avoids harm. We do not foresee any direct negative societal impacts resulting from

this work; however, as with any advancement in machine learning, we acknowledge the potential for dual-use applications. We encourage the community to apply this technology responsibly and ethically.

## REPRODUCIBILITY STATEMENT

Our implementation is built upon the JAX framework. The code will be made available upon acceptance. All experiments were conducted on a single NVIDIA A100 GPU. The datasets used, D4RL and OGBench, are publicly available benchmarks. Comprehensive details regarding network architectures, hyperparameters, and specific environment configurations are provided in Appendix A, which we believe are sufficient for the community to verify our results and build upon this work.

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

## A  IMPLEMENTATION DETAILS

### A.1  HYPERPARAMETERS AND ARCHITECTURE

Across all experiments, we set the learning rates for the critic network $Z_\phi$ and $Z_{\tilde{\phi}}$ to be $3 \times 10^{-4}$ and $1 \times 10^{-4}$. We use a sample size of $M = 51$ for the distributional estimation (following the classical setting migrated from Bellemare et al. (2017)), an exploration constant of $\delta = 0.3$, a Huber loss parameter of $\zeta = 1$, and 10 integration steps for the flow models. For all neural network components, which are realized as $[512, 512, 512, 512]$-sized multi-layer perceptions (MLPs), we follow the architectural configuration of FQL. To ensure a fair and direct comparison, we also adopt the identical hyperparameters for the actor, including the behavioral cloning coefficient $\alpha_{\text{BC}}$, as specified in FQL. We provide the complete list of hyperparameters in Table 2.

Due to computational resource constraints, our hyperparameter selection largely adheres to the configurations established in FQL. A comprehensive list of general hyperparameters is provided in Table 2, while task-specific settings are detailed in Table 3. For key baseline methods, we made the following specific adjustments:

- **For IQL and IFQL**, we set exploring expectile value to be 0.9, while the AWR inverse temperature, $\alpha$, is tuned on a per-environment basis (see Table 3).

- **For ReBRAC**, the actor BC coefficient, $\alpha_1$, and the critic BC coefficient, $\alpha_2$, are set on a per-environment basis, as specified in Table 3.

- **For IDQL**, the expectile value, $\tau$, is configured to 0.7 for OGBench locomotion and Adroit tasks, and to 0.9 for OGBench manipulation and AntMaze tasks. The number of action samples, $N$, is also tuned individually per task (see Table 3). Consistent with its original protocol, the agent is trained for 3 million steps (1.5 million for the value function).

- **For CAC**, the Q-loss coefficient, $\eta$, is determined based on the environment-specific values listed in Table 3.

Table 2: **Hyperparameters for DFC.** ($*$) Denotes hyperparameters for which the values are kept identical to the FQL configuration to ensure a fair comparison.

| Hyperparameter | Value |
|---|---|
| Distributional critic learning rate | 0.0001 |
| Main critic learning rate | 0.0003 |
| Distributional sample size | 50 |
| Exploration constant $\delta$ | 0.3 |
| Huber loss parameter $\zeta$ | 1 |
| Actor learning rate ($*$) | 0.0003 |
| Optimizer ($*$) | Adam (Adam et al., 2015) |
| Gradient steps ($*$) | 1000000 (default), 500000 (Offline D4RL, pixel-based OGBench) |
| Minibatch size ($*$) | 256 |
| MLP dimensions ($*$) | $[512, 512, 512, 512]$ |
| Nonlinearity ($*$) | GELU (Hendrycks & Gimpel, 2016) |
| Target network smoothing coefficient ($*$) | 0.005 |
| Discount factor $\gamma$ ($*$) | 0.99 (default), 0.995 (`antmaze-giant`, `humanoidmaze`, `antsoccer`) |
| Image augmentation probability ($*$) | 0.5 |
| Flow steps | 10 |
| Flow time sampling distribution | $\mathcal{U}([0, 1])$ |
| BC coefficient $\alpha$ ($*$) | See Table 3. |

For the state-based OGBench tasks, DFC is trained for one million gradient steps, whereas for the D4RL and pixel-based OGBench tasks, training is conducted for 500K steps, in alignment with the FQL protocol. The agent's performance is evaluated every 100K steps over 50 episodes. For the offline-to-online experiments, we report the performance metrics at both one million and two million total steps. For reporting final scores on OGBench, we follow the protocol of FQL and average the success rates over the last three evaluation epochs (i.e., 800K, 900K, and 1M steps for state-based tasks; 300K, 400K, and 500K for pixel-based tasks). For D4RL, consistent with Tarasov et al. (2023b), we report the performance at the final training epoch.

In accordance with the official implementation of OGBench, for all pixel-based experiments, we use a compact version of the IMPALA encoder (Espeholt et al., 2018) with random random-shift augmentation.

Table 3: **Task-specific hyperparameters.** For a detailed description of each hyperparameter, we refer the reader to Appendix A.1. Some task-specific hyperparameter values are adopted from FQL. To ensure experimental consistency, a single hyperparameter configuration is applied uniformly to all five tasks derived from each OGBench environment. An em-dash "-" indicates that a corresponding result is not available.

| Task | IQL $\alpha$ | ReBRAC $(\alpha_1, \alpha_2)$ | IDQL $N$ | CAC $\eta$ | IFQL $N$ | DFC (FQL) $\alpha$ |
|---|---|---|---|---|---|---|
| antmaze-large-navigate-singletask-v0 | 10 | $(0.003, 0.01)$ | 32 | 1 | 32 | 10 |
| antmaze-giant-navigate-singletask-v0 | 10 | $(0.003, 0.01)$ | 32 | 1 | 32 | 10 |
| humanoidmaze-medium-navigate-singletask-v0 | 10 | $(0.01, 0.01)$ | 32 | 0.03 | 32 | 30 |
| humanoidmaze-large-navigate-singletask-v0 | 10 | $(0.01, 0.01)$ | 32 | 1 | 32 | 30 |
| antsoccer-arena-navigate-singletask-v0 | 1 | $(0.01, 0.01)$ | 32 | 1 | 64 | 10 |
| cube-single-play-singletask-v0 | 1 | $(1, 0)$ | 32 | 0.003 | 32 | 300 |
| cube-double-play-singletask-v0 | 0.3 | $(0.1, 0)$ | 32 | 0.3 | 32 | 300 |
| scene-play-singletask-v0 | 10 | $(0.1, 0.01)$ | 32 | 0.3 | 32 | 300 |
| puzzle-3x3-play-singletask-v0 | 10 | $(0.3, 0.01)$ | 32 | 0.01 | 32 | 1000 |
| puzzle-4x4-play-singletask-v0 | 3 | $(0.3, 0.01)$ | 32 | 0.01 | 32 | 1000 |
| antmaze-umaze-v2 | 10 | $(0.003, 0.002)$ | 32 | 0.01 | 32 | 10 |
| antmaze-umaze-diverse-v2 | 10 | $(0.003, 0.001)$ | 32 | 0.01 | 32 | 10 |
| antmaze-medium-play-v2 | 10 | $(0.001, 0.0005)$ | 32 | 0.01 | 32 | 10 |
| antmaze-medium-diverse-v2 | 10 | $(0.001, 0.0)$ | 32 | 0.01 | 32 | 10 |
| antmaze-large-play-v2 | 10 | $(0.002, 0.001)$ | 32 | 4.5 | 32 | 3 |
| antmaze-large-diverse-v2 | 10 | $(0.002, 0.002)$ | 32 | 3.5 | 32 | 3 |
| pen-human-v1 | 3 | $(0.1, 0.5)$ | 32 | 0.003 | 32 | 10000 |
| pen-cloned-v1 | 3 | $(0.05, 0.5)$ | 32 | 0.003 | 32 | 10000 |
| pen-expert-v1 | 3 | $(0.01, 0.01)$ | 32 | 0.03 | 32 | 3000 |
| door-human-v1 | 3 | $(0.1, 0.1)$ | 32 | 0.03 | 32 | 30000 |
| door-cloned-v1 | 3 | $(0.01, 0.1)$ | 32 | 0.03 | 128 | 30000 |
| door-expert-v1 | 3 | $(0.05, 0.01)$ | 32 | 0.03 | 32 | 30000 |
| hammer-human-v1 | 3 | $(0.01, 0.5)$ | 128 | 0.03 | 32 | 30000 |
| hammer-cloned-v1 | 3 | $(0.1, 0.5)$ | 32 | 0.003 | 32 | 10000 |
| hammer-expert-v1 | 3 | $(0.01, 0.01)$ | 32 | 0.03 | 32 | 30000 |
| relocate-human-v1 | 3 | $(0.1, 0.01)$ | 32 | 0.01 | 128 | 10000 |
| relocate-cloned-v1 | 3 | $(0.1, 0.01)$ | 64 | 0.01 | 32 | 30000 |
| relocate-expert-v1 | 3 | $(0.05, 0.01)$ | 32 | 0.003 | 32 | 30000 |
| visual-cube-single-play-singletask-task1-v0 | 1 | $(1, 0)$ | - | - | 32 | 300 |
| visual-cube-double-play-singletask-task1-v0 | 0.3 | $(0.1, 0)$ | - | - | 32 | 100 |
| visual-scene-play-singletask-task1-v0 | 10 | $(0.1, 0.01)$ | - | - | 32 | 100 |
| visual-puzzle-3x3-play-singletask-task1-v0 | 10 | $(0.3, 0.01)$ | - | - | 32 | 300 |
| visual-puzzle-4x4-play-singletask-task1-v0 | 3 | $(0.3, 0.01)$ | - | - | 32 | 300 |

## A.2 DATASETS AND ENVIRONMENTS

Our empirical evaluation is conducted on two prominent offline reinforcement learning benchmarks: OGBench and D4RL. These benchmarks provide a diverse set of environments and tasks designed to test the limits of offline RL algorithms, particularly their ability to stitch together suboptimal trajectories into effective policies. Detailed results are provided in Table 4.

**OGBench** (Park et al., 2025a) features a semi-sparse reward structure where the reward is defined as the negative count of remaining subtasks required to achieve a fixed goal. This structure presents distinct challenges: locomotion tasks typically involve a single subtask (reaching a goal) with binary rewards (0 or $-1$), while manipulation tasks can involve up to 16 sequential subtasks. The datasets are collected from policies executing random tasks, resulting in highly suboptimal data that necessitates strong stitching capabilities from the learning algorithm.

Our experiments leverage the following OGBench environments, categorized by input type and task domain:

- **State-Based Tasks (50 tasks from 10 environments):** These tasks require control based on proprioceptive states.
  - *Locomotion*: The agent, either a quadruped (ant) or a humanoid, must navigate through various maze layouts or dribble a ball to a target location.
    - antmaze-large-navigate-v0
    - antmaze-giant-navigate-v0

- `humanoidmaze-medium-navigate-v0`
- `humanoidmaze-large-navigate-v0`
- `antsoccer-arena-navigate-v0`

  - *Manipulation*: A robotic arm must manipulate diverse objects. These tasks range from simple object interaction (`cube`) to long-horizon, multi-object control (`scene`) and challenges requiring combinatorial generalization (`puzzle`).
    - `cube-single-play-v0`
    - `cube-double-play-v0`
    - `scene-play-v0`
    - `puzzle-3x3-play-v0`
    - `puzzle-4x4-play-v0`

- **Pixel-Based Tasks (5 tasks from 5 environments):** These are the visual counterparts to the manipulation tasks, requiring control solely from $64 \times 64 \times 3$ image observations.

  - `visual-cube-single-play-v0`
  - `visual-cube-double-play-v0`
  - `visual-scene-play-v0`
  - `visual-puzzle-3x3-play-v0`
  - `visual-puzzle-4x4-play-v0`

**D4RL** (Fu et al., 2020) is also evaluated on 18 challenging tasks to facilitate direct comparison with a broad range of prior work. These tasks are selected to cover complex locomotion and dexterous manipulation challenges.

- **AntMaze Tasks (6 tasks):** These locomotion tasks share a similar high-level objective with their OGBench counterparts but feature different maze layouts and dataset characteristics.

  - `antmaze-umaze-v2`
  - `antmaze-umaze-diverse-v2`
  - `antmaze-medium-play-v2`
  - `antmaze-medium-diverse-v2`
  - `antmaze-large-play-v2`
  - `antmaze-large-diverse-v2`

- **Adroit Tasks (12 tasks):** These tasks demand dexterous manipulation using a high-dimensional (24-DoF) robotic hand, with objectives such as spinning a pen, opening a door, hammering a nail, and relocating an object.

  - `pen-human-v1, -cloned-v1, -expert-v1`
  - `door-human-v1, -cloned-v1, -expert-v1`
  - `hammer-human-v1, -cloned-v1, -expert-v1`
  - `relocate-human-v1, -cloned-v1, -expert-v1`

Evaluation metrics adhere to the standard protocols for each benchmark: success rates for OGBench and D4RL AntMaze tasks, and normalized returns for Adroit tasks.

## B    DETAILED RESULTS

In this section, we present detailed results for DFC, including comprehensive empirical analyses of the critic and actor architectures, training time, and sample efficiency.

**Full results.** Our full results are shown in Table 4 and Table 5. Learning curves for selected D4RL and OGBench tasks are presented in Figure 3, clearly illustrating the differences in stability and performance across methods.

**Return Distribution Visualization.** We visualize the return distribution of C51, IQN and DFC in Figure 5. C51 predicts a distribution looking like impulses around support points, IQN often produces overly multi-modal distributions, while DFC generates distributions that exhibit distinct

multimodality and align much more closely with the ground-truth return distribution. Numerically, our method performs 2 times better than the best baselines under 1-Wasserstein metric.

We also evaluated the absolute error between the predicted scalar mean and the ground-truth mean across the three tasks. For completeness, we include the vanilla FQL baseline that directly regresses to the mean value. Our method (DFC) achieves a mean absolute error of 0.0432, compared to 0.1085 for FQL and 0.1684 for IQN. These results demonstrate that DFC substantially outperforms the baselines, achieving a normalized mean absolute error that is less than half that of the strongest competing method.

**Run time comparison.** We present the time consumption comparison in Figure 4. DFC is only slightly slower than FQL in training speeds, while not being slower than most other methods such as IQL, FQL, FBRAC and IFQL in inference speeds.

**Distributional methods.** We have provided a comprehensive evaluation of DFC against established distributional approaches, C51 and IQN, as summarized in Table 6. Using identical actor architectures across all methods, DFC consistently surpasses both C51 and IQN on D4RL adroit tasks and OGBench state-based benchmarks, achieving an average improvement of approximately $14\%$. These results highlight the effectiveness and generality of our approach.

**Ensemble methods.** We conducted experiments comparing our approach to ensembled critic baselines in Table 7, evaluating both 2-ensemble and 4-ensemble Q-networks of FQL. Across diverse environments—including D4RL's adroit tasks and OGBench state-based benchmarks—our DFC method consistently delivers the strongest performance, outperforming ensemble-based critics in nearly all cases. These results demonstrate that DFC achieves clear advantages over traditional ensemble methods, confirming the effectiveness and robustness of our distributional critic design.

**Non-exclusiveness to flow policy.** We present ablation experiments in Table 8, which evaluate our model's performance across different policy architectures. The results show that, our method consistently outperforms the vanilla critic baseline on both locomotion and manipulation tasks on OGBench benchmarks. This demonstrates the robustness of our approach to variations in the policy model and underscores the advantage of using a distributional critic in diverse environments.

**Sample numbers.** Ablations of the number of quantiles $M$ are presented in Table 9. Following standard practice (Bellemare et al., 2017), we vary $M$ in $\{51, 101\}$ and evaluate performance both on D4RL adroit manipulation tasks and OGBench state-based tasks. The results indicate that the optimal choice of $M$ can vary between environments: neither value consistently dominates, and there is no clear pattern favoring larger or smaller $M$ across all tasks. This suggests that the impact of $M$ is nuanced and may depend on specific environment dynamics rather than a universal rule.

## C    USAGE OF LARGE LANGUAGE MODELS (LLMS)

In the final stages of preparing this manuscript, a large language model (LLM) was utilized solely for the purpose of language polishing and editing. The core scientific ideas, experimental design, results, and initial drafts were entirely the work of the human authors. The LLM's role was confined to improving grammar, refining sentence structure, and ensuring stylistic consistency. All suggestions provided by the LLM were critically reviewed and manually approved by the authors, who retain full responsibility for the entirety of the paper's content, in accordance with the ICLR Code of Ethics.

## D    PROOF OF PROPOSITION 1

*Proof.* We define the space of distributional value functions as $\mathcal{Z}$, mapping state-action pairs to probability measures over returns. To analyze convergence, we equip $\mathcal{Z}$ with the *maximal 1-Wasserstein metric* Bellemare et al. (2017), denoted as $\bar{d}_\infty$, which is defined as the supremum of the 1-Wasserstein distance over the state-action space:

$$\bar{d}_\infty(Z_1, Z_2) := \sup_{s \in \mathcal{S}, a \in \mathcal{A}} W_1(Z_1(s, a), Z_2(s, a)). \tag{9}$$

Our proof relies on decomposing the combined operator $\hat{\mathcal{T}}$ into two sequential components: the distributional Bellman operator $\mathcal{T}^\pi$ and the quantile projection operator $\Pi_{W_1}$. We analyze the Lipschitz properties under the maximal 1-Wasserstein metric $\bar{d}_\infty$.

**Contraction of the Bellman Operator** First, we recall the property of the distributional Bellman operator $\mathcal{T}^\pi$, defined as $Z(s,a) \leftarrow R(s,a) + \gamma Z(s', a')$. As established in Lemma 3 of Bellemare et al. (2017), this operator is a $\gamma$-contraction in $\bar{d}_\infty$. Specifically, for any two distributions $Z_1, Z_2$:

$$\bar{d}_\infty(\mathcal{T}^\pi Z_1, \mathcal{T}^\pi Z_2) \leq \gamma \bar{d}_\infty(Z_1, Z_2). \tag{10}$$

his contraction property stems directly from the scaling homogeneity of the Wasserstein metric, where $W_1(\gamma U, \gamma V) = \gamma W_1(U, V)$. However, since our framework incorporates an additional distillation step, it is necessary to verify whether this intermediate projection affects the contraction property of the overall process.

Our objective is to establish the contraction of the composite operator $\Pi_{W_1} \mathcal{T}^\pi$. Invoking Proposition 2 of (Dabney et al., 2018b), we obtain:

$$\bar{d}_\infty(\Pi_{W_1}(\mathcal{T}^\pi Z_1), \Pi_{W_1}(\mathcal{T}^\pi Z_2)) \leq \gamma \bar{d}_\infty(Z_1, Z_2). \tag{11}$$

Since $\gamma \in [0, 1)$, $\hat{\mathcal{T}}$ is a contraction mapping on the complete metric space $(\mathcal{Z}, \bar{d}_\infty)$. By the Banach Fixed-Point Theorem, the sequence $Z_{k+1} \leftarrow \hat{\mathcal{T}} Z_k$ converges to a unique fixed point. $\square$

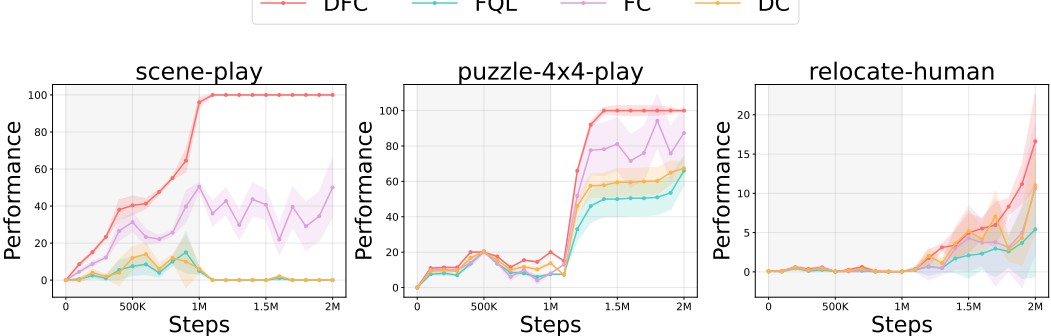

Figure 3: **Learning curves for offline-to-online RL.** Conducted on the D4RL and OGBench benchmark suites. The initial one million steps, conducted in the offline setting, are highlighted by the gray shaded area.

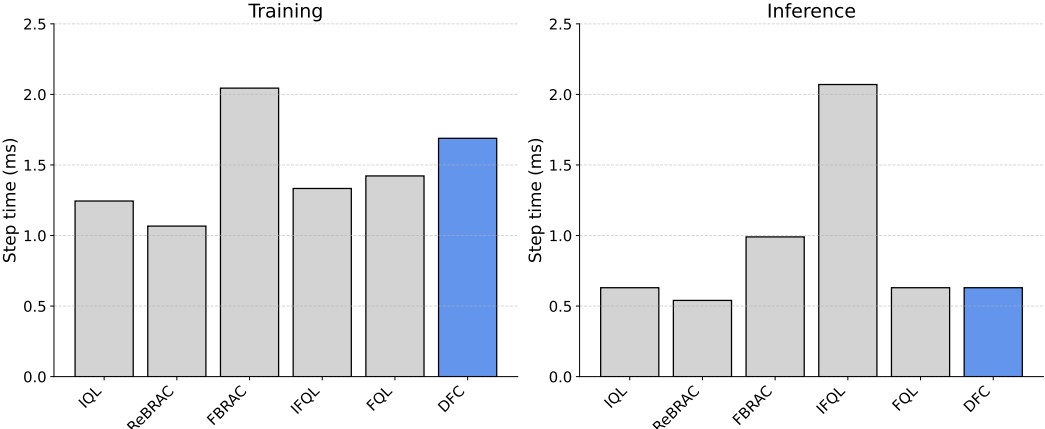

Figure 4: **Run time comparison.** DFC is only slightly slower than FQL in training speeds, while not being slower than most other methods in inference speeds.

Table 4: **Offline RL results.** The following tables present a comprehensive evaluation of our method against key baselines across a suite of over 70 tasks from the OGBench and D4RL benchmarks. To ensure statistical robustness, all experiments were conducted with multiple random seeds: 8 for state-based tasks and 4 for pixel-based tasks. The best scores are highlighted in **bold** and the suboptimal scores are labeled as underlined. The asterisk (∗) denotes the default task within each environment group.

| Task | IQL | ReBRAC | IDQL | CAC | IFQL | FQL | DFC |
|---|---|---|---|---|---|---|---|
| antmaze-large-navigate-singletask-task1-v0 (∗) | 48 ±9 | **91** ±10 | 0 ±0 | 42 ±7 | 24 ±17 | 80 ±8 | 90 ±1 |
| antmaze-large-navigate-singletask-task2-v0 | 42 ±6 | **88** ±4 | 14 ±8 | 1 ±1 | 8 ±3 | 57 ±10 | 74 ±4 |
| antmaze-large-navigate-singletask-task3-v0 | 72 ±7 | 51 ±18 | 26 ±8 | 49 ±10 | 52 ±17 | 93 ±3 | **98** ±1 |
| antmaze-large-navigate-singletask-task4-v0 | 51 ±9 | 84 ±7 | 62 ±25 | 17 ±6 | 18 ±8 | 80 ±4 | **89** ±2 |
| antmaze-large-navigate-singletask-task5-v0 | 54 ±22 | 90 ±2 | 2 ±2 | 55 ±6 | 38 ±18 | 83 ±4 | **91** ±3 |
| antmaze-giant-navigate-singletask-task1-v0 (∗) | 0 ±0 | **27** ±22 | 0 ±0 | 0 ±0 | 0 ±0 | 4 ±5 | 25 ±12 |
| antmaze-giant-navigate-singletask-task2-v0 | 1 ±1 | **16** ±17 | 0 ±0 | 0 ±0 | 0 ±0 | 9 ±7 | 1 ±1 |
| antmaze-giant-navigate-singletask-task3-v0 | 0 ±0 | **34** ±22 | 0 ±0 | 0 ±0 | 0 ±0 | 0 ±1 | 1 ±1 |
| antmaze-giant-navigate-singletask-task4-v0 | 0 ±0 | 5 ±12 | 0 ±0 | 0 ±0 | 0 ±0 | 14 ±23 | **40** ±45 |
| antmaze-giant-navigate-singletask-task5-v0 | 19 ±7 | **49** ±22 | 0 ±1 | 0 ±0 | 13 ±9 | 16 ±28 | 30 ±10 |
| humanoidmaze-medium-navigate-singletask-task1-v0 (∗) | 32 ±7 | 16 ±9 | 1 ±1 | 38 ±19 | **69** ±19 | 19 ±12 | 35 ±9 |
| humanoidmaze-medium-navigate-singletask-task2-v0 | 41 ±9 | 18 ±16 | 1 ±1 | 47 ±35 | 85 ±11 | 94 ±3 | **99** ±1 |
| humanoidmaze-medium-navigate-singletask-task3-v0 | 25 ±5 | 36 ±13 | 0 ±1 | 83 ±18 | 49 ±49 | 74 ±18 | **96** ±3 |
| humanoidmaze-medium-navigate-singletask-task4-v0 | 0 ±1 | **15** ±16 | 1 ±1 | 5 ±4 | 1 ±1 | 3 ±4 | 3 ±1 |
| humanoidmaze-medium-navigate-singletask-task5-v0 | 66 ±4 | 24 ±20 | 1 ±1 | 91 ±5 | 98 ±2 | 97 ±2 | **100** ±1 |
| humanoidmaze-large-navigate-singletask-task1-v0 (∗) | 3 ±1 | 2 ±1 | 0 ±0 | 1 ±1 | 6 ±2 | 7 ±6 | **15** ±6 |
| humanoidmaze-large-navigate-singletask-task2-v0 | **0** ±0 | **0** ±0 | **0** ±0 | **0** ±0 | **0** ±0 | **0** ±0 | **0** ±0 |
| humanoidmaze-large-navigate-singletask-task3-v0 | 7 ±3 | 8 ±4 | 3 ±1 | 2 ±3 | **48** ±10 | 11 ±7 | 19 ±5 |
| humanoidmaze-large-navigate-singletask-task4-v0 | 1 ±0 | 1 ±1 | 0 ±0 | 0 ±1 | 1 ±1 | 2 ±3 | **4** ±2 |
| humanoidmaze-large-navigate-singletask-task5-v0 | 1 ±1 | **2** ±2 | 0 ±0 | 0 ±0 | 0 ±0 | 1 ±3 | 1 ±2 |
| antsoccer-arena-navigate-singletask-task1-v0 | 14 ±5 | 0 ±0 | 44 ±12 | 1 ±3 | 61 ±25 | 77 ±4 | **84** ±3 |
| antsoccer-arena-navigate-singletask-task2-v0 | 17 ±7 | 0 ±1 | 15 ±12 | 0 ±0 | 75 ±3 | 88 ±3 | **96** ±3 |
| antsoccer-arena-navigate-singletask-task3-v0 | 6 ±4 | 0 ±0 | 0 ±0 | 8 ±19 | 14 ±22 | 61 ±6 | **69** ±5 |
| antsoccer-arena-navigate-singletask-task4-v0 (∗) | 3 ±2 | 0 ±0 | 0 ±1 | 0 ±0 | 16 ±9 | 39 ±6 | **52** ±7 |
| antsoccer-arena-navigate-singletask-task5-v0 | 2 ±2 | 0 ±0 | 0 ±0 | 0 ±0 | 0 ±1 | 36 ±9 | **66** ±5 |
| cube-single-play-singletask-task1-v0 | 88 ±3 | 89 ±5 | 95 ±2 | 77 ±28 | 79 ±4 | 97 ±2 | **100** ±0 |
| cube-single-play-singletask-task2-v0 (∗) | 85 ±8 | 92 ±4 | 96 ±2 | 80 ±30 | 73 ±3 | 97 ±2 | **100** ±0 |
| cube-single-play-singletask-task3-v0 | 91 ±5 | 93 ±3 | 99 ±1 | 98 ±1 | 88 ±4 | 98 ±2 | **100** ±0 |
| cube-single-play-singletask-task4-v0 | 73 ±6 | 92 ±3 | 93 ±4 | 91 ±2 | 79 ±6 | 94 ±3 | **99** ±1 |
| cube-single-play-singletask-task5-v0 | 78 ±9 | 87 ±8 | 90 ±6 | 80 ±20 | 77 ±7 | 93 ±3 | **100** ±0 |
| cube-double-play-singletask-task1-v0 | 27 ±5 | 45 ±6 | 39 ±19 | 21 ±8 | 35 ±9 | 61 ±9 | **84** ±4 |
| cube-double-play-singletask-task2-v0 (∗) | 1 ±1 | 7 ±3 | 16 ±10 | 2 ±2 | 9 ±5 | 36 ±6 | **49** ±2 |
| cube-double-play-singletask-task3-v0 | 0 ±0 | 4 ±1 | 17 ±8 | 3 ±1 | 8 ±5 | 22 ±5 | **31** ±1 |
| cube-double-play-singletask-task4-v0 | 0 ±0 | 1 ±1 | 0 ±1 | 0 ±1 | 1 ±1 | 5 ±2 | **11** ±4 |
| cube-double-play-singletask-task5-v0 | 4 ±3 | 4 ±2 | 1 ±1 | 3 ±2 | 17 ±6 | 19 ±10 | **28** ±11 |
| scene-play-singletask-task1-v0 | 94 ±3 | 95 ±2 | 100 ±0 | 100 ±1 | 98 ±3 | 100 ±0 | 100 ±0 |
| scene-play-singletask-task2-v0 (∗) | 12 ±3 | 50 ±13 | 33 ±14 | 50 ±40 | 0 ±0 | 76 ±9 | **96** ±3 |
| scene-play-singletask-task3-v0 | 32 ±7 | 55 ±16 | 94 ±4 | 49 ±16 | 54 ±19 | 98 ±1 | **100** ±0 |
| scene-play-singletask-task4-v0 | 0 ±1 | 3 ±3 | 4 ±3 | 0 ±0 | 0 ±0 | 5 ±1 | **20** ±9 |
| scene-play-singletask-task5-v0 | **0** ±0 | **0** ±0 | **0** ±0 | **0** ±0 | **0** ±0 | **0** ±0 | **0** ±0 |
| puzzle-3x3-play-singletask-task1-v0 | 33 ±6 | **97** ±4 | 52 ±12 | **97** ±2 | 94 ±3 | 90 ±4 | **97** ±1 |
| puzzle-3x3-play-singletask-task2-v0 | 4 ±3 | 1 ±1 | 0 ±1 | 0 ±0 | 1 ±2 | 16 ±5 | **36** ±1 |
| puzzle-3x3-play-singletask-task3-v0 | 3 ±2 | 3 ±1 | 0 ±0 | 0 ±0 | 0 ±0 | 10 ±3 | **17** ±3 |
| puzzle-3x3-play-singletask-task4-v0 (∗) | 2 ±1 | 2 ±1 | 0 ±0 | 0 ±0 | 0 ±0 | **16** ±5 | 21 ±5 |
| puzzle-3x3-play-singletask-task5-v0 | 3 ±2 | 5 ±3 | 0 ±0 | 0 ±0 | 0 ±0 | 16 ±3 | **29** ±5 |
| puzzle-4x4-play-singletask-task1-v0 | 12 ±2 | 26 ±4 | 48 ±5 | 44 ±10 | 49 ±9 | 34 ±8 | **68** ±5 |
| puzzle-4x4-play-singletask-task2-v0 | 7 ±4 | 12 ±4 | 14 ±5 | 0 ±0 | 4 ±4 | 16 ±5 | **24** ±2 |
| puzzle-4x4-play-singletask-task3-v0 | 9 ±3 | 15 ±3 | 34 ±5 | 29 ±12 | 50 ±14 | 18 ±5 | 49 ±3 |
| puzzle-4x4-play-singletask-task4-v0 (∗) | 5 ±2 | 10 ±3 | **26** ±6 | 1 ±1 | 21 ±11 | 11 ±3 | 20 ±3 |
| puzzle-4x4-play-singletask-task5-v0 | 4 ±1 | 7 ±3 | **24** ±11 | 0 ±0 | 2 ±2 | 7 ±3 | 14 ±4 |
| antmaze-umaze-v2 | 77 | 98 | 94 | 66 ±5 | 92 ±6 | 96 ±2 | **100** ±0 |
| antmaze-umaze-diverse-v2 | 54 | 84 | 80 | 66 ±11 | 62 ±12 | 89 ±5 | **96** ±0 |
| antmaze-medium-play-v2 | 66 | **90** | 84 | 49 ±24 | 56 ±15 | 78 ±7 | 85 ±3 |
| antmaze-medium-diverse-v2 | 74 | 84 | **85** | 0 ±1 | 60 ±25 | 71 ±13 | 85 ±3 |
| antmaze-large-play-v2 | 42 | 52 | 64 | 0 ±0 | 55 ±9 | 84 ±7 | **91** ±1 |
| antmaze-large-diverse-v2 | 30 | 64 | 68 | 0 ±0 | 64 ±8 | 83 ±4 | **95** ±1 |
| pen-human-v1 | 78 | **103** | 76 ±10 | 64 ±8 | 71 ±12 | 53 ±6 | 61 ±7 |
| pen-cloned-v1 | 83 | **103** | 64 ±7 | 56 ±10 | 80 ±11 | 74 ±11 | 90 ±7 |
| pen-expert-v1 | 128 | **152** | 140 ±6 | 103 ±9 | 139 ±5 | 142 ±6 | 147 ±1 |
| door-human-v1 | 3 | −0 | 6 ±2 | 5 ±2 | **7** ±2 | 0 ±0 | 6 ±3 |
| door-cloned-v1 | 3 | 0 | 0 ±0 | 1 ±0 | 2 ±2 | 2 ±1 | **6** ±5 |
| door-expert-v1 | **107** | 106 | 105 ±1 | 98 ±4 | 104 ±2 | 104 ±1 | 105 ±0 |
| hammer-human-v1 | 2 | 0 | 2 ±1 | 2 ±0 | **3** ±1 | 1 ±1 | 4 ±1 |
| hammer-cloned-v1 | 2 | 5 | 2 ±1 | 1 ±1 | 2 ±1 | 11 ±9 | **14** ±13 |
| hammer-expert-v1 | 129 | **134** | 125 ±4 | 92 ±11 | 117 ±9 | 125 ±3 | 128 ±0 |
| relocate-human-v1 | **0** | **0** | **0** ±0 | **0** ±0 | **0** ±0 | **0** ±0 | ⊬ ±0 |
| relocate-cloned-v1 | 0 | **2** | −0 ±0 | −0 ±0 | −0 ±0 | −0 ±0 | 1 ±0 |
| relocate-expert-v1 | 106 | **108** | 107 ±1 | 93 ±6 | 104 ±3 | 107 ±1 | 109 ±0 |
| visual-cube-single-play-singletask-task1-v0 | 70 ±12 | 83 ±6 | - | - | 49 ±7 | 81 ±12 | **85** ±10 |
| visual-cube-double-play-singletask-task1-v0 | **34** ±23 | 4 ±4 | - | - | 8 ±6 | 21 ±11 | 23 ±10 |
| visual-scene-play-singletask-task1-v0 | 97 ±2 | 98 ±4 | - | - | 86 ±10 | 98 ±3 | **100** ±0 |
| visual-puzzle-3x3-play-singletask-task1-v0 | 7 ±15 | 88 ±4 | - | - | **100** ±0 | 94 ±1 | 95 ±7 |
| visual-puzzle-4x4-play-singletask-task1-v0 | 0 ±0 | 26 ±6 | - | - | 8 ±15 | 33 ±6 | **37** ±6 |

Table 5: **Ablation study of the DFC model on a single-layer critic architecture.** We compare the full model against its key components: a Flow-only Critic (FC) and a Distribchaoutional-only Critic (DC). FQL is included for baseline performance reference. The meaning of $\rightarrow$ illustrates the performance improvement during the offline-to-online scheme. The performance to the left of $\rightarrow$ is the result under offline setting while that to the right is the result after 100k epochs of online fine-tuning. The asterisk ($*$) denotes the default task within each environment group.

| Task | FQL | FC | DC | DFC |
|---|---|---|---|---|
| antmaze-large-navigate-singletask-task1-v0 ($*$) | $80_{\pm8} \rightarrow 100_{\pm0}$ | $86_{\pm3} \rightarrow 100_{\pm0}$ | $88_{\pm1} \rightarrow 100_{\pm0}$ | $90_{\pm1} \rightarrow 100_{\pm0}$ |
| antmaze-large-navigate-singletask-task2-v0 | $57_{\pm10} \rightarrow 88_{\pm2}$ | $72_{\pm16} \rightarrow 88_{\pm3}$ | $69_{\pm8} \rightarrow 88_{\pm1}$ | $74_{\pm4} \rightarrow 89_{\pm1}$ |
| antmaze-large-navigate-singletask-task3-v0 | $93_{\pm3} \rightarrow 100_{\pm0}$ | $96_{\pm6} \rightarrow 100_{\pm0}$ | $96_{\pm2} \rightarrow 100_{\pm0}$ | $98_{\pm1} \rightarrow 100_{\pm0}$ |
| antmaze-large-navigate-singletask-task4-v0 | $80_{\pm4} \rightarrow 97_{\pm1}$ | $86_{\pm4} \rightarrow 98_{\pm1}$ | $88_{\pm2} \rightarrow 98_{\pm0}$ | $89_{\pm2} \rightarrow 99_{\pm1}$ |
| antmaze-large-navigate-singletask-task5-v0 | $83_{\pm4} \rightarrow 99_{\pm1}$ | $90_{\pm6} \rightarrow 100_{\pm1}$ | $91_{\pm3} \rightarrow 100_{\pm1}$ | $91_{\pm3} \rightarrow 100_{\pm1}$ |
| antmaze-giant-navigate-singletask-task1-v0 ($*$) | $4_{\pm5} \rightarrow 97_{\pm2}$ | $16_{\pm21} \rightarrow 98_{\pm3}$ | $22_{\pm12} \rightarrow 94_{\pm1}$ | $25_{\pm12} \rightarrow 98_{\pm2}$ |
| antmaze-giant-navigate-singletask-task2-v0 | $9_{\pm7} \rightarrow 98_{\pm0}$ | $0_{\pm1} \rightarrow 98_{\pm0}$ | $1_{\pm1} \rightarrow 100_{\pm0}$ | $1_{\pm1} \rightarrow 99_{\pm1}$ |
| antmaze-giant-navigate-singletask-task3-v0 | $0_{\pm1} \rightarrow 0_{\pm0}$ | $0_{\pm0} \rightarrow 0_{\pm0}$ | $1_{\pm0} \rightarrow 3_{\pm4}$ | $1_{\pm1} \rightarrow 10_{\pm3}$ |
| antmaze-giant-navigate-singletask-task4-v0 | $14_{\pm23} \rightarrow 95_{\pm1}$ | $38_{\pm54} \rightarrow 94_{\pm1}$ | $39_{\pm44} \rightarrow 94_{\pm1}$ | $40_{\pm45} \rightarrow 99_{\pm1}$ |
| antmaze-giant-navigate-singletask-task5-v0 | $16_{\pm28} \rightarrow 100_{\pm0}$ | $0_{\pm6} \rightarrow 100_{\pm0}$ | $16_{\pm7} \rightarrow 100_{\pm0}$ | $30_{\pm10} \rightarrow 100_{\pm0}$ |
| humanoidmaze-medium-navigate-singletask-task1-v0 ($*$) | $19_{\pm12} \rightarrow 22_{\pm13}$ | $40_{\pm17} \rightarrow 0_{\pm16}$ | $34_{\pm9} \rightarrow 34_{\pm11}$ | $35_{\pm9} \rightarrow 59_{\pm32}$ |
| humanoidmaze-medium-navigate-singletask-task2-v0 | $94_{\pm3} \rightarrow 99_{\pm1}$ | $98_{\pm3} \rightarrow 98_{\pm1}$ | $98_{\pm1} \rightarrow 98_{\pm1}$ | $99_{\pm1} \rightarrow 100_{\pm0}$ |
| humanoidmaze-medium-navigate-singletask-task3-v0 | $74_{\pm18} \rightarrow 79_{\pm16}$ | $98_{\pm24} \rightarrow 0_{\pm20}$ | $90_{\pm10} \rightarrow 90_{\pm49}$ | $96_{\pm3} \rightarrow 99_{\pm1}$ |
| humanoidmaze-medium-navigate-singletask-task4-v0 | $3_{\pm4} \rightarrow 13_{\pm19}$ | $2_{\pm1} \rightarrow 4_{\pm24}$ | $2_{\pm1} \rightarrow 34_{\pm3}$ | $3_{\pm1} \rightarrow 66_{\pm24}$ |
| humanoidmaze-medium-navigate-singletask-task5-v0 | $97_{\pm2} \rightarrow 77_{\pm40}$ | $100_{\pm3} \rightarrow 100_{\pm49}$ | $99_{\pm1} \rightarrow 99_{\pm0}$ | $100_{\pm1} \rightarrow 100_{\pm0}$ |
| humanoidmaze-large-navigate-singletask-task1-v0 ($*$) | $7_{\pm6} \rightarrow 0_{\pm0}$ | $0_{\pm6} \rightarrow 0_{\pm0}$ | $9_{\pm5} \rightarrow 9_{\pm10}$ | $15_{\pm6} \rightarrow 16_{\pm3}$ |
| humanoidmaze-large-navigate-singletask-task2-v0 | $0_{\pm0} \rightarrow 0_{\pm0}$ | $0_{\pm0} \rightarrow 0_{\pm0}$ | $0_{\pm0} \rightarrow 0_{\pm0}$ | $0_{\pm0} \rightarrow 0_{\pm0}$ |
| humanoidmaze-large-navigate-singletask-task3-v0 | $11_{\pm7} \rightarrow 26_{\pm35}$ | $18_{\pm16} \rightarrow 12_{\pm47}$ | $16_{\pm7} \rightarrow 32_{\pm1}$ | $19_{\pm5} \rightarrow 67_{\pm3}$ |
| humanoidmaze-large-navigate-singletask-task4-v0 | $2_{\pm3} \rightarrow 0_{\pm0}$ | $0_{\pm0} \rightarrow 0_{\pm0}$ | $2_{\pm1} \rightarrow 5_{\pm3}$ | $4_{\pm2} \rightarrow 20_{\pm14}$ |
| humanoidmaze-large-navigate-singletask-task5-v0 | $1_{\pm3} \rightarrow 0_{\pm0}$ | $0_{\pm0} \rightarrow 0_{\pm0}$ | $1_{\pm1} \rightarrow 1_{\pm0}$ | $1_{\pm2} \rightarrow 3_{\pm2}$ |
| antsoccer-arena-navigate-singletask-task1-v0 | $77_{\pm4} \rightarrow 99_{\pm1}$ | $74_{\pm16} \rightarrow 98_{\pm1}$ | $80_{\pm6} \rightarrow 100_{\pm1}$ | $84_{\pm3} \rightarrow 100_{\pm1}$ |
| antsoccer-arena-navigate-singletask-task2-v0 | $88_{\pm3} \rightarrow 95_{\pm2}$ | $90_{\pm7} \rightarrow 94_{\pm3}$ | $92_{\pm3} \rightarrow 94_{\pm1}$ | $96_{\pm3} \rightarrow 100_{\pm0}$ |
| antsoccer-arena-navigate-singletask-task3-v0 | $61_{\pm6} \rightarrow 91_{\pm3}$ | $60_{\pm18} \rightarrow 90_{\pm4}$ | $65_{\pm8} \rightarrow 90_{\pm1}$ | $69_{\pm5} \rightarrow 94_{\pm2}$ |
| antsoccer-arena-navigate-singletask-task4-v0 ($*$) | $39_{\pm6} \rightarrow 71_{\pm4}$ | $50_{\pm21} \rightarrow 70_{\pm6}$ | $50_{\pm9} \rightarrow 68_{\pm1}$ | $52_{\pm7} \rightarrow 82_{\pm4}$ |
| antsoccer-arena-navigate-singletask-task5-v0 | $36_{\pm9} \rightarrow 69_{\pm5}$ | $50_{\pm8} \rightarrow 72_{\pm6}$ | $57_{\pm5} \rightarrow 72_{\pm8}$ | $66_{\pm5} \rightarrow 86_{\pm1}$ |
| cube-single-play-singletask-task1-v0 | $97_{\pm2} \rightarrow 100_{\pm0}$ | $98_{\pm7} \rightarrow 100_{\pm0}$ | $98_{\pm2} \rightarrow 100_{\pm0}$ | $100_{\pm0} \rightarrow 100_{\pm0}$ |
| cube-single-play-singletask-task2-v0 ($*$) | $97_{\pm2} \rightarrow 100_{\pm0}$ | $100_{\pm1} \rightarrow 100_{\pm0}$ | $100_{\pm0} \rightarrow 100_{\pm0}$ | $100_{\pm0} \rightarrow 100_{\pm0}$ |
| cube-single-play-singletask-task3-v0 | $98_{\pm2} \rightarrow 100_{\pm0}$ | $98_{\pm1} \rightarrow 100_{\pm0}$ | $100_{\pm0} \rightarrow 100_{\pm0}$ | $100_{\pm0} \rightarrow 100_{\pm0}$ |
| cube-single-play-singletask-task4-v0 | $94_{\pm3} \rightarrow 100_{\pm0}$ | $96_{\pm4} \rightarrow 100_{\pm0}$ | $97_{\pm1} \rightarrow 100_{\pm0}$ | $99_{\pm1} \rightarrow 100_{\pm0}$ |
| cube-single-play-singletask-task5-v0 | $93_{\pm3} \rightarrow 99_{\pm1}$ | $94_{\pm4} \rightarrow 100_{\pm1}$ | $97_{\pm1} \rightarrow 98_{\pm0}$ | $100_{\pm0} \rightarrow 100_{\pm0}$ |
| cube-double-play-singletask-task1-v0 | $61_{\pm9} \rightarrow 97_{\pm3}$ | $78_{\pm20} \rightarrow 96_{\pm4}$ | $80_{\pm8} \rightarrow 96_{\pm0}$ | $84_{\pm4} \rightarrow 100_{\pm0}$ |
| cube-double-play-singletask-task2-v0 ($*$) | $36_{\pm6} \rightarrow 95_{\pm2}$ | $50_{\pm6} \rightarrow 94_{\pm3}$ | $49_{\pm3} \rightarrow 94_{\pm1}$ | $49_{\pm2} \rightarrow 98_{\pm2}$ |
| cube-double-play-singletask-task3-v0 | $22_{\pm5} \rightarrow 95_{\pm3}$ | $26_{\pm7} \rightarrow 94_{\pm4}$ | $29_{\pm3} \rightarrow 98_{\pm0}$ | $31_{\pm1} \rightarrow 100_{\pm1}$ |
| cube-double-play-singletask-task4-v0 | $5_{\pm2} \rightarrow 29_{\pm23}$ | $8_{\pm4} \rightarrow 32_{\pm33}$ | $9_{\pm3} \rightarrow 19_{\pm23}$ | $11_{\pm4} \rightarrow 34_{\pm16}$ |
| cube-double-play-singletask-task5-v0 | $19_{\pm10} \rightarrow 97_{\pm4}$ | $24_{\pm18} \rightarrow 98_{\pm6}$ | $28_{\pm11} \rightarrow 96_{\pm2}$ | $28_{\pm11} \rightarrow 100_{\pm1}$ |
| scene-play-singletask-task1-v0 | $100_{\pm0} \rightarrow 100_{\pm0}$ | $100_{\pm1} \rightarrow 100_{\pm0}$ | $100_{\pm1} \rightarrow 100_{\pm0}$ | $100_{\pm0} \rightarrow 100_{\pm0}$ |
| scene-play-singletask-task2-v0 ($*$) | $76_{\pm9} \rightarrow 100_{\pm0}$ | $92_{\pm6} \rightarrow 100_{\pm0}$ | $94_{\pm4} \rightarrow 96_{\pm1}$ | $96_{\pm3} \rightarrow 100_{\pm0}$ |
| scene-play-singletask-task3-v0 | $98_{\pm1} \rightarrow 100_{\pm0}$ | $100_{\pm3} \rightarrow 100_{\pm0}$ | $99_{\pm1} \rightarrow 100_{\pm0}$ | $100_{\pm0} \rightarrow 100_{\pm0}$ |
| scene-play-singletask-task4-v0 | $5_{\pm1} \rightarrow 0_{\pm0}$ | $6_{\pm3} \rightarrow 0_{\pm0}$ | $12_{\pm6} \rightarrow 13_{\pm8}$ | $20_{\pm9} \rightarrow 20_{\pm12}$ |
| scene-play-singletask-task5-v0 | $0_{\pm0} \rightarrow 0_{\pm0}$ | $0_{\pm0} \rightarrow 0_{\pm0}$ | $0_{\pm0} \rightarrow 0_{\pm0}$ | $0_{\pm0} \rightarrow 0_{\pm0}$ |
| puzzle-3x3-play-singletask-task1-v0 | $90_{\pm4} \rightarrow 97_{\pm2}$ | $90_{\pm13} \rightarrow 100_{\pm0}$ | $91_{\pm5} \rightarrow 100_{\pm0}$ | $97_{\pm1} \rightarrow 100_{\pm0}$ |
| puzzle-3x3-play-singletask-task2-v0 | $16_{\pm5} \rightarrow 0_{\pm0}$ | $8_{\pm4} \rightarrow 0_{\pm0}$ | $21_{\pm2} \rightarrow 21_{\pm25}$ | $36_{\pm1} \rightarrow 37_{\pm1}$ |
| puzzle-3x3-play-singletask-task3-v0 | $10_{\pm3} \rightarrow 0_{\pm0}$ | $8_{\pm6} \rightarrow 0_{\pm0}$ | $12_{\pm4} \rightarrow 12_{\pm10}$ | $17_{\pm3} \rightarrow 16_{\pm1}$ |
| puzzle-3x3-play-singletask-task4-v0 ($*$) | $16_{\pm5} \rightarrow 0_{\pm0}$ | $14_{\pm6} \rightarrow 0_{\pm0}$ | $16_{\pm5} \rightarrow 17_{\pm13}$ | $21_{\pm5} \rightarrow 22_{\pm5}$ |
| puzzle-3x3-play-singletask-task5-v0 | $16_{\pm3} \rightarrow 0_{\pm0}$ | $20_{\pm7} \rightarrow 0_{\pm0}$ | $23_{\pm5} \rightarrow 23_{\pm17}$ | $29_{\pm5} \rightarrow 30_{\pm6}$ |
| puzzle-4x4-play-singletask-task1-v0 | $34_{\pm8} \rightarrow 100_{\pm0}$ | $20_{\pm7} \rightarrow 100_{\pm0}$ | $44_{\pm5} \rightarrow 100_{\pm0}$ | $68_{\pm5} \rightarrow 100_{\pm0}$ |
| puzzle-4x4-play-singletask-task2-v0 | $16_{\pm5} \rightarrow 0_{\pm0}$ | $14_{\pm3} \rightarrow 0_{\pm0}$ | $18_{\pm2} \rightarrow 18_{\pm14}$ | $24_{\pm2} \rightarrow 24_{\pm3}$ |
| puzzle-4x4-play-singletask-task3-v0 | $18_{\pm5} \rightarrow 100_{\pm0}$ | $16_{\pm8} \rightarrow 100_{\pm0}$ | $31_{\pm5} \rightarrow 100_{\pm0}$ | $49_{\pm3} \rightarrow 100_{\pm0}$ |
| puzzle-4x4-play-singletask-task4-v0 ($*$) | $11_{\pm3} \rightarrow 53_{\pm50}$ | $8_{\pm1} \rightarrow 60_{\pm71}$ | $14_{\pm2} \rightarrow 60_{\pm47}$ | $20_{\pm3} \rightarrow 100_{\pm0}$ |
| puzzle-4x4-play-singletask-task5-v0 | $7_{\pm3} \rightarrow 0_{\pm0}$ | $4_{\pm3} \rightarrow 0_{\pm0}$ | $9_{\pm3} \rightarrow 9_{\pm7}$ | $14_{\pm4} \rightarrow 13_{\pm3}$ |
| antmaze-umaze-v2 | $96_{\pm2} \rightarrow 99_{\pm1}$ | $98_{\pm4} \rightarrow 100_{\pm1}$ | $99_{\pm2} \rightarrow 98_{\pm0}$ | $100_{\pm0} \rightarrow 100_{\pm0}$ |
| antmaze-umaze-diverse-v2 | $89_{\pm5} \rightarrow 100_{\pm0}$ | $80_{\pm8} \rightarrow 100_{\pm0}$ | $90_{\pm4} \rightarrow 100_{\pm0}$ | $96_{\pm0} \rightarrow 100_{\pm0}$ |
| antmaze-medium-play-v2 | $78_{\pm7} \rightarrow 98_{\pm2}$ | $74_{\pm14} \rightarrow 98_{\pm3}$ | $78_{\pm5} \rightarrow 98_{\pm0}$ | $85_{\pm3} \rightarrow 100_{\pm0}$ |
| antmaze-medium-diverse-v2 | $71_{\pm13} \rightarrow 95_{\pm1}$ | $70_{\pm8} \rightarrow 96_{\pm1}$ | $76_{\pm4} \rightarrow 96_{\pm0}$ | $85_{\pm3} \rightarrow 99_{\pm2}$ |
| antmaze-large-play-v2 | $84_{\pm7} \rightarrow 93_{\pm6}$ | $86_{\pm7} \rightarrow 94_{\pm8}$ | $89_{\pm4} \rightarrow 86_{\pm0}$ | $91_{\pm1} \rightarrow 98_{\pm0}$ |
| antmaze-large-diverse-v2 | $83_{\pm4} \rightarrow 89_{\pm2}$ | $86_{\pm3} \rightarrow 88_{\pm3}$ | $90_{\pm2} \rightarrow 88_{\pm3}$ | $95_{\pm1} \rightarrow 97_{\pm1}$ |
| pen-human-v1 | $53_{\pm6} \rightarrow 134_{\pm2}$ | $50_{\pm21} \rightarrow 135_{\pm3}$ | $58_{\pm12} \rightarrow 135_{\pm2}$ | $61_{\pm7} \rightarrow 135_{\pm2}$ |
| pen-cloned-v1 | $74_{\pm11} \rightarrow 142_{\pm8}$ | $81_{\pm13} \rightarrow 143_{\pm11}$ | $82_{\pm7} \rightarrow 134_{\pm8}$ | $90_{\pm7} \rightarrow 144_{\pm6}$ |
| pen-expert-v1 | $142_{\pm6} \rightarrow 158_{\pm3}$ | $142_{\pm9} \rightarrow 157_{\pm5}$ | $144_{\pm3} \rightarrow 155_{\pm2}$ | $147_{\pm1} \rightarrow 161_{\pm1}$ |
| door-human-v1 | $0_{\pm0} \rightarrow 83_{\pm1}$ | $0_{\pm0} \rightarrow 83_{\pm1}$ | $4_{\pm2} \rightarrow 83_{\pm3}$ | $6_{\pm3} \rightarrow 82_{\pm2}$ |
| door-cloned-v1 | $2_{\pm1} \rightarrow 78_{\pm20}$ | $2_{\pm2} \rightarrow 69_{\pm27}$ | $6_{\pm2} \rightarrow 64_{\pm23}$ | $6_{\pm5} \rightarrow 82_{\pm22}$ |
| door-expert-v1 | $104_{\pm1} \rightarrow 106_{\pm0}$ | $105_{\pm2} \rightarrow 106_{\pm0}$ | $105_{\pm1} \rightarrow 106_{\pm0}$ | $105_{\pm0} \rightarrow 106_{\pm0}$ |
| hammer-human-v1 | $1_{\pm1} \rightarrow 114_{\pm5}$ | $1_{\pm2} \rightarrow 112_{\pm6}$ | $2_{\pm1} \rightarrow 112_{\pm6}$ | $4_{\pm1} \rightarrow 121_{\pm8}$ |
| hammer-cloned-v1 | $11_{\pm9} \rightarrow 129_{\pm8}$ | $3_{\pm7} \rightarrow 126_{\pm11}$ | $12_{\pm11} \rightarrow 126_{\pm9}$ | $14_{\pm13} \rightarrow 130_{\pm7}$ |
| hammer-expert-v1 | $125_{\pm3} \rightarrow 132_{\pm0}$ | $126_{\pm4} \rightarrow 132_{\pm0}$ | $127_{\pm1} \rightarrow 132_{\pm0}$ | $128_{\pm0} \rightarrow 133_{\pm0}$ |
| relocate-human-v1 | $0_{\pm0} \rightarrow 10_{\pm8}$ | $0_{\pm0} \rightarrow 11_{\pm11}$ | $0_{\pm0} \rightarrow 11_{\pm10}$ | $0_{\pm0} \rightarrow 10_{\pm6}$ |
| relocate-cloned-v1 | $-0_{\pm0} \rightarrow 31_{\pm27}$ | $0_{\pm0} \rightarrow 44_{\pm35}$ | $1_{\pm0} \rightarrow 28_{\pm33}$ | $1_{\pm0} \rightarrow 36_{\pm19}$ |
| relocate-expert-v1 | $107_{\pm1} \rightarrow 106_{\pm1}$ | $107_{\pm2} \rightarrow 107_{\pm2}$ | $108_{\pm2} \rightarrow 108_{\pm3}$ | $109_{\pm0} \rightarrow 110_{\pm0}$ |
| visual-cube-single-play-singletask-task1-v0 | $81_{\pm12}$ | $84_{\pm11}$ | $81_{\pm8}$ | $85_{\pm10}$ |
| visual-cube-double-play-singletask-task1-v0 | $21_{\pm11}$ | $22_{\pm21}$ | $20_{\pm15}$ | $23_{\pm10}$ |
| visual-scene-play-singletask-task1-v0 | $98_{\pm3}$ | $100_{\pm1}$ | $99_{\pm1}$ | $100_{\pm0}$ |
| visual-puzzle-3x3-play-singletask-task1-v0 | $94_{\pm1}$ | $90_{\pm8}$ | $90_{\pm6}$ | $95_{\pm7}$ |
| visual-puzzle-4x4-play-singletask-task1-v0 | $33_{\pm6}$ | $26_{\pm21}$ | $24_{\pm15}$ | $37_{\pm6}$ |

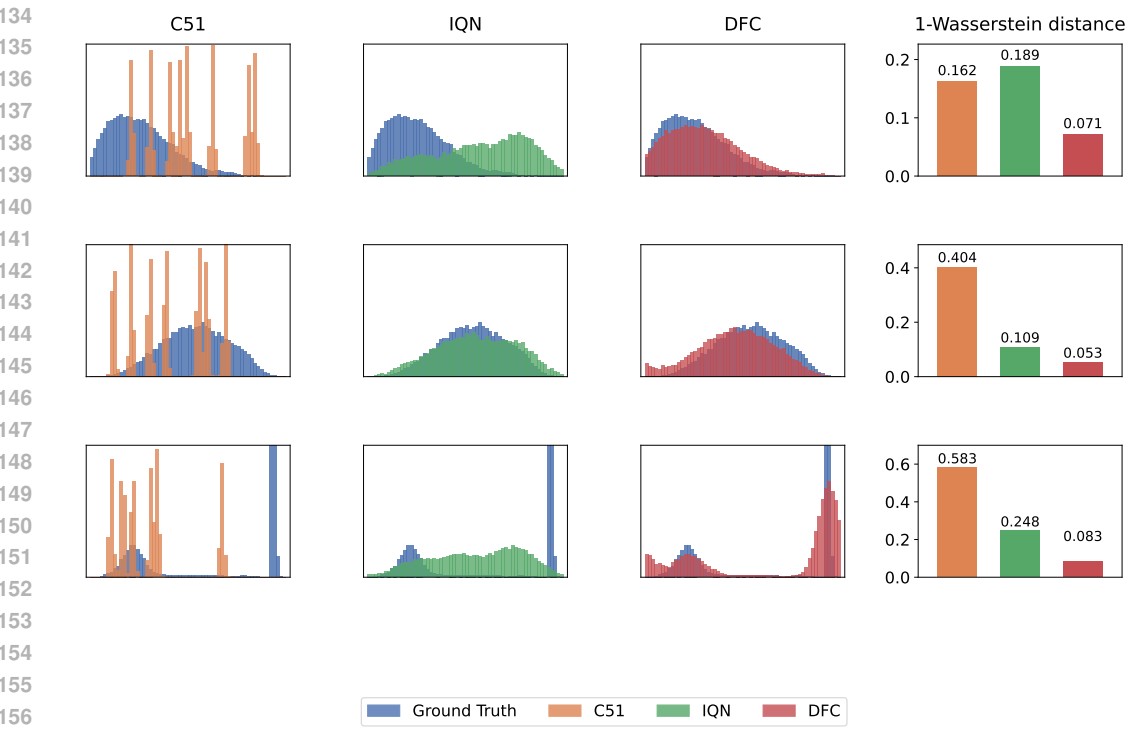

Figure 5: **Return distribution visulization.** We visualize the return distribution of C51, IQN and DFC. C51 predicts a distribution looking like impulses around support points, IQN often produces overly multimodal distributions, while DFC generates distributions that exhibit distinct multimodality and align much more closely with the ground-truth return distribution.

Table 6: **Ablation study of distributional methods.** We compare the DFC model against other distributional methods: C51 and IQN. We all use the same flow-actor structure according to FQL.

| env_name | C51 | IQN | DFC |
|---|---|---|---|
| pen-human-v1 | $\mathbf{69 \pm 8}$ | $\mathbf{69 \pm 3}$ | $61 \pm 7$ |
| pen-cloned-v1 | $67 \pm 9$ | $80 \pm 11$ | $\mathbf{90 \pm 7}$ |
| pen-expert-v1 | $110 \pm 3$ | $118 \pm 19$ | $\mathbf{147 \pm 1}$ |
| door-human-v1 | $0 \pm 0$ | $0 \pm 0$ | $\mathbf{6 \pm 3}$ |
| door-cloned-v1 | $0 \pm 0$ | $0 \pm 0$ | $\mathbf{6 \pm 5}$ |
| door-expert-v1 | $104 \pm 1$ | $\mathbf{105 \pm 0}$ | $\mathbf{105 \pm 0}$ |
| hammer-human-v1 | $3 \pm 1$ | $2 \pm 1$ | $\mathbf{4 \pm 1}$ |
| hammer-cloned-v1 | $0 \pm 0$ | $0 \pm 0$ | $\mathbf{14 \pm 13}$ |
| hammer-expert-v1 | $122 \pm 1$ | $121 \pm 7$ | $\mathbf{128 \pm 0}$ |
| relocate-human-v1 | $0 \pm 0$ | $0 \pm 0$ | $0 \pm 0$ |
| relocate-cloned-v1 | $0 \pm 0$ | $0 \pm 0$ | $\mathbf{1 \pm 0}$ |
| relocate-expert-v1 | $103 \pm 0$ | $103 \pm 0$ | $\mathbf{109 \pm 0}$ |
| antmaze-large-navigate-singletask-v0 | $1 \pm 1$ | $30 \pm 8$ | $\mathbf{90 \pm 1}$ |
| antmaze-giant-navigate-singletask-v0 | $0 \pm 0$ | $0 \pm 0$ | $\mathbf{25 \pm 12}$ |
| humanoidmaze-medium-navigate-singletask-v0 | $2 \pm 4$ | $33 \pm 4$ | $\mathbf{35 \pm 9}$ |
| humanoidmaze-large-navigate-singletask-v0 | $0 \pm 0$ | $8 \pm 6$ | $\mathbf{14 \pm 6}$ |
| antsoccer-arena-navigate-singletask-v0 | $2 \pm 2$ | $27 \pm 6$ | $\mathbf{52 \pm 7}$ |
| cube-single-play-singletask-v0 | $4 \pm 2$ | $98 \pm 6$ | $\mathbf{100 \pm 0}$ |
| cube-double-play-singletask-v0 | $0 \pm 0$ | $24 \pm 9$ | $\mathbf{49 \pm 2}$ |
| scene-play-singletask-v0 | $0 \pm 0$ | $1 \pm 0$ | $\mathbf{96 \pm 3}$ |
| puzzle-3x3-play-singletask-v0 | $1 \pm 4$ | $0 \pm 0$ | $\mathbf{21 \pm 5}$ |
| puzzle-4x4-play-singletask-v0 | $0 \pm 0$ | $\mathbf{23 \pm 2}$ | $20 \pm 3$ |

Table 7: **Ablation study of ensemble methods.** We compare the DFC model against FQL with ensembled value models. DFC performs better on all the tasks listed.

| Task | DFC | FQL ($\#Q = 2$) | FQL ($\#Q = 4$) |
|---|---|---|---|
| pen-human-v1 | $\mathbf{61 \pm 7}$ | $53 \pm 6$ | $39 \pm 12$ |
| pen-cloned-v1 | $\mathbf{90 \pm 7}$ | $74 \pm 11$ | $76 \pm 15$ |
| pen-expert-v1 | $\mathbf{147 \pm 1}$ | $142 \pm 6$ | $145 \pm 2$ |
| door-human-v1 | $\mathbf{6 \pm 3}$ | $6 \pm 3$ | $0 \pm 0$ |
| door-cloned-v1 | $\mathbf{6 \pm 5}$ | $2 \pm 1$ | $1 \pm 1$ |
| door-expert-v1 | $\mathbf{105 \pm 0}$ | $104 \pm 1$ | $105 \pm 0$ |
| hammer-human-v1 | $\mathbf{4 \pm 1}$ | $3 \pm 1$ | $1 \pm 1$ |
| hammer-cloned-v1 | $\mathbf{14 \pm 13}$ | $11 \pm 9$ | $4 \pm 1$ |
| hammer-expert-v1 | $\mathbf{128 \pm 0}$ | $125 \pm 3$ | $127 \pm 1$ |
| relocate-human-v1 | $\mathbf{0 \pm 0}$ | $0 \pm 0$ | $0 \pm 0$ |
| relocate-cloned-v1 | $\mathbf{1 \pm 0}$ | $-0 \pm 0$ | $-0 \pm 0$ |
| relocate-expert-v1 | $\mathbf{109 \pm 0}$ | $107 \pm 1$ | $109 \pm 1$ |
| antmaze-large-navigate-singletask-v0 | $\mathbf{90 \pm 1}$ | $80 \pm 8$ | $56 \pm 12$ |
| antmaze-giant-navigate-singletask-v0 | $\mathbf{25 \pm 12}$ | $4 \pm 5$ | $17 \pm 4$ |
| humanoidmaze-medium-navigate-singletask-v0 | $\mathbf{35 \pm 9}$ | $19 \pm 12$ | $22 \pm 14$ |
| humanoidmaze-large-navigate-singletask-v0 | $\mathbf{14 \pm 6}$ | $7 \pm 6$ | $6 \pm 5$ |
| antsoccer-arena-navigate-singletask-v0 | $\mathbf{52 \pm 7}$ | $39 \pm 6$ | $47 \pm 6$ |
| cube-single-play-singletask-v0 | $\mathbf{100 \pm 0}$ | $97 \pm 2$ | $98 \pm 2$ |
| cube-double-play-singletask-v0 | $\mathbf{49 \pm 2}$ | $36 \pm 6$ | $46 \pm 3$ |
| scene-play-singletask-v0 | $\mathbf{96 \pm 3}$ | $76 \pm 9$ | $83 \pm 4$ |
| puzzle-3x3-play-singletask-v0 | $\mathbf{21 \pm 5}$ | $16 \pm 5$ | $19 \pm 5$ |
| puzzle-4x4-play-singletask-v0 | $\mathbf{20 \pm 3}$ | $11 \pm 3$ | $6 \pm 3$ |

Table 8: **Ablation study of critic structure.** We plugin the DF critic to ReBRAC and IQL. The results show that, our method consistently outperforms the vanilla critic baseline on both locomotion and manipulation tasks on OGBench benchmarks.

| Task | ReBRAC | | IQL | |
|---|---|---|---|---|
| | Vanilla | DF Critic | Vanilla | DF Critic |
| antmaze-large-navigate-singletask-v0 | $91 \pm 10$ | $91 \pm 1$ | $48 \pm 9$ | $67 \pm 4$ |
| antmaze-giant-navigate-singletask-v0 | $27 \pm 22$ | $74 \pm 5$ | $0 \pm 0$ | $0 \pm 0$ |
| humanoidmaze-medium-navigate-singletask-v0 | $16 \pm 9$ | $23 \pm 1$ | $32 \pm 7$ | $43 \pm 3$ |
| humanoidmaze-large-navigate-singletask-v0 | $2 \pm 1$ | $5 \pm 10$ | $3 \pm 1$ | $5 \pm 1$ |
| antsoccer-arena-navigate-singletask-v0 | $0 \pm 0$ | $23 \pm 10$ | $3 \pm 2$ | $14 \pm 1$ |
| cube-single-play-singletask-v0 | $92 \pm 4$ | $100 \pm 0$ | $85 \pm 8$ | $100 \pm 0$ |
| cube-double-play-singletask-v0 | $7 \pm 3$ | $8 \pm 0$ | $1 \pm 1$ | $4 \pm 1$ |
| scene-play-singletask-v0 | $50 \pm 13$ | $99 \pm 1$ | $12 \pm 3$ | $64 \pm 10$ |
| puzzle-3x3-play-singletask-v0 | $2 \pm 1$ | $98 \pm 0$ | $2 \pm 1$ | $4 \pm 1$ |
| puzzle-4x4-play-singletask-v0 | $10 \pm 3$ | $3 \pm 10$ | $5 \pm 2$ | $11 \pm 2$ |

Table 9: **Ablation study of sample numbers** $M$**.** We ablate $M$ in $\{51, 101\}$ following Bellemare et al. (2017). The best scores are highlighted in **bold**.

| Task | $M = 51$ | $M = 101$ |
|---|---|---|
| pen-human-v1 | $61 \pm 7$ | $\mathbf{84 \pm 3}$ |
| pen-cloned-v1 | $\mathbf{90 \pm 7}$ | $77 \pm 5$ |
| pen-expert-v1 | $\mathbf{147 \pm 1}$ | $130 \pm 12$ |
| door-human-v1 | $6 \pm 3$ | $\mathbf{8 \pm 2}$ |
| door-cloned-v1 | $\mathbf{6 \pm 5}$ | $1 \pm 1$ |
| door-expert-v1 | $105 \pm 0$ | $\mathbf{106 \pm 0}$ |
| hammer-human-v1 | $4 \pm 1$ | $3 \pm 1$ |
| hammer-cloned-v1 | $\mathbf{14 \pm 13}$ | $1 \pm 1$ |
| hammer-expert-v1 | $\mathbf{128 \pm 0}$ | $123 \pm 1$ |
| relocate-human-v1 | $\mathbf{0 \pm 0}$ | $\mathbf{0 \pm 0}$ |
| relocate-cloned-v1 | $\mathbf{1 \pm 0}$ | $0 \pm 0$ |
| relocate-expert-v1 | $\mathbf{109 \pm 0}$ | $107 \pm 1$ |
| antmaze-large-navigate-singletask-v0 | $90 \pm 1$ | $\mathbf{93 \pm 2}$ |
| antmaze-giant-navigate-singletask-v0 | $\mathbf{25 \pm 12}$ | $20 \pm 6$ |
| humanoidmaze-medium-navigate-singletask-v0 | $\mathbf{35 \pm 9}$ | $33 \pm 8$ |
| humanoidmaze-large-navigate-singletask-v0 | $14 \pm 6$ | $14 \pm 5$ |
| antsoccer-arena-navigate-singletask-v0 | $\mathbf{52 \pm 7}$ | $50 \pm 3$ |
| cube-single-play-singletask-v0 | $100 \pm 0$ | $100 \pm 0$ |
| cube-double-play-singletask-v0 | $49 \pm 2$ | $\mathbf{74 \pm 8}$ |
| scene-play-singletask-v0 | $96 \pm 3$ | $\mathbf{99 \pm 1}$ |
| puzzle-3x3-play-singletask-v0 | $21 \pm 5$ | $\mathbf{52 \pm 3}$ |
| puzzle-4x4-play-singletask-v0 | $\mathbf{20 \pm 3}$ | $18 \pm 5$ |
| cube-double-play-singletask-v0 | $49 \pm 2$ | $\mathbf{74 \pm 8}$ |
| scene-play-singletask-v0 | $96 \pm 3$ | $\mathbf{99 \pm 1}$ |
| puzzle-3x3-play-singletask-v0 | $21 \pm 5$ | $\mathbf{52 \pm 3}$ |
| puzzle-4x4-play-singletask-v0 | $\mathbf{20 \pm 3}$ | $18 \pm 5$ |

