# OpenReview forum: "Unleashing Flow Policies with Distributional Critics"
_ICLR.cc/2026/Conference — Submitted to ICLR 2026_

### Official Review · Reviewer_S7BU · 2025-10-25

**Soundness:** 3
**Presentation:** 3
**Contribution:** 3
**Rating:** 4
**Confidence:** 4

**Summary:**

This paper seeks to enhance the expressiveness of value functions by learning value distributions instead of relying on expected scalar values. To achieve this, the paper employs diffusion models/flow matching to optimize self-bootstrapped value distributions, as these methods are well-suited for modeling complex distributions without parametric restrictions. However, once the value distributions are obtained, extracting the policy by maximizing these distributions becomes challenging due to the instability and inefficiency of optimizing through the full flow/diffusion chain. To address this, the authors apply distillation techniques along with quantile regression to transform the multi-step flow/diffusion ODEs/SDEs into a one-step noise-to-quantile distribution mapping. This simplification allows for efficient maximization of the distilled expected values to derive the policy. Experimental results on D4RL and OGBench demonstrate the strong potential of distributional RL and highlight the benefits of using diffusion/flow models to capture the risk and uncertainty inherent in value estimations.

**Strengths:**

1. Using diffusion/flow to model the value distributions is novel. Recently, there are also some methods that do the similar things, but they are concurrent works.
2. The proposed methods seem easy to optimize, since all components are optimized in a decoupled manner, therefore has few limitations on training instability.
3. The empirical performance on D4RL and OGBench are strong, demonstrating the promises of combining distirbutional RL and flow/diffusions.
4. The presentation is good, easy to follow.

**Weaknesses:**

1. The authors have already obtained a distilled value distribution but still use its mean value to derive the policy, as shown in Section 3.3. This approach undermines the benefits of value distribution modeling, as the mean value could be directly regressed without incurring the additional computational costs.  It's necessary to provide a solid discussion as well as more empirical supports to explain why directly regressing to the mean value is suboptimal.

2. Since the authors have already obtained the entire value distribution, will it bring additional benefits by not using the mean value, but considering other properties of distributions?

3. I don't know if the self-bootstrapped value distribution learning in Eq (4-5) is stable and whether it requires additional tricks to stablize the training. Providing more details to optimize Eq (4-5) or showing some empirical experiments to visualize the learned value distirbution would be good and interesting to address this concern.

4. Figure 1 can be further beautified.

Currently, I have no additional major concerns. The manuscript is relatively strong, presenting an approach that is easy to optimize, the idea is relatively novel, the empirical results are strong. However, more empirical evidence or discussion is needed to address the points raised above. I will reconsider the score after the rebuttal.

**Questions:**

Please see weaknesses for details.

---

> ### Author Response · Authors · 2025-11-22
> **Weakness 1 and 2**
>
> ## W1: “The authors have already obtained a distilled value distribution...”.
> We appreciate this thoughtful question regarding the use of the expected value for the actor update. We would like to first clarify that this design choice aligns with the standard protocol in continuous Distributional RL, and then use an intuitive example to explain why the distributional *learning process* remains critical even when the *update target* is a scalar.
>
> **1. Alignment with Established Standards**
> Using the expectation of the learned distribution to update the policy is the established standard in continuous Actor-Critic Distributional RL.
> * **Standard Practice:** Seminal works such as **D4PG** [1] explicitly calculate the policy gradient using the expectation of the value distribution.
> * **Objective Consistency:** This practice ensures that the agent optimizes the standard RL objective-maximizing the expected cumulative return, while leveraging distributional learning to improve the *quality* and *stability* of the value estimation itself[2].
>
> **2. Stability in Multimodal Environments**
> The reviewer asks if properties like multimodality are lost. We argue that explicitly modeling multimodality is exactly what allows us to retrieve a stable mean.
> * **Scalar Critic Failure:** In a highly stochastic or multimodal environment (e.g., a 50% chance of 0 reward and 50% chance of 100), a scalar critic often oscillates or fails to converge to the true mean (50).
> * **Distributional Critic Success:** Our critic explicitly learns the two modes (0 and 100). Because the distribution is modeled correctly, the derived expectation is mathematically precise and stable.
>
> We provide concrete evidence (beyond above toy example) in Figure 5 in our new pdf, where we visualize the ground truth distribution and  learned return distributions of C51, IQN, and DFC. The results verify that the return landscape in some scenario is indeed **multimodal**. Crucially, DFC achieves a **significantly superior fit** (quantitively over 2.5 times measured by the lowest 1-Wasserstein distance) compared to C51 and IQN. This confirms that capturing these complex distributional details is practically achievable via our method and superior to baseline distributional approaches. Please check it.
>
> [1] Gabriel Barth-Maron , Matthew W. Hoffman, David Budden, Will Dabney, Dan Horgan, Dhruva TB, Alistair Muldal, Nicolas Heess, Timothy Lillicrap. Distributed Distributional Deterministic Policy Gradients. In ICLR, 2018.
>
> [2] Marc G. Bellemare, Will Dabney, and R´emi Munos. A distributional perspective on reinforcement learning. In ICML, 2017.
>
> ## W2: “Since the authors have already obtained the entire value distribution...”.
>
> We thank the reviewer for this insightful suggestion. We fully agree that having access to the full value distribution opens up significant possibilities beyond simple expectation maximization.
>
> **1. Potential for Risk-Sensitive RL**
> The reviewer is correct that we could leverage other properties, such as the tail risk (CVaR) or variance.
> * **Risk-Aversion:** In safety-critical applications, optimizing for the Conditional Value-at-Risk (CVaR) instead of the mean can produce policies that avoid catastrophic failures, even if it sacrifices some average performance.
>
> **2. Justification for Using the Mean in This Work**
> While these extensions are promising, we chose to use the mean for the actor update in this paper for the following reasons:  The standard metrics for D4RL and OGBench (Success Rate and Normalized Score) are designed to evaluate the **Expected Return**. Mathematically, maximizing the mean of the distribution is the direct objective for these benchmarks. Optimizing for risk (e.g., CVaR) often leads to conservative behaviors that might lower the "Average Score," making it difficult to compare fairly against state-of-the-art baselines on these specific leaderboards.
>
> Therefore, We view the use of non-mean properties as a compelling direction for future work, particularly for tasks explicitly focusing on **Safety RL** or **Risk-Sensitive Control**. For the general performance benchmarks targeted in this study, using the high-fidelity distribution to derive a **more accurate mean** provided the most significant gains.

---

> ### Author Response · Authors · 2025-11-22
> **Weakness 3 and 4**
>
> ## W3: "I don't know if the self-bootstrapped value distribution learning in Eq (4-5) is stable and...  empirical experiments to visualize the learned value distirbution would be good and interesting to address this concern".
>
>  We have plotted the learned value distribution against ground truth and other distributional methods in Figure 5 of Appendix B in our new pdf.  DFC achieves a **significantly superior fit** (quantitively over 2.5 times measured by the lowest 1-Wasserstein distance) compared to C51 and IQN. This confirms that capturing these complex distributional details is practically achievable via our method and superior to baseline distributional approaches.
>
> We show that the combination of the distillation projection induced by quantile regression with the Bellman operator is a contraction, guaranteeing convergence. For further details see Proposition 1 in our new pdf.
>
> ## W4:  "Beautification of Figure 1.".
>
> Thank you for your suggestion. We have updated it in our new pdf. Please check it. To improve clarity, we have redesigned the module layout so the structure of each module is more distinct. We have updated the vector icons and carefully selected new bounding box styles, aiming for a more visually coherent presentation. We hope these changes address your feedback and enhance the overall readability.

---

> > ### Comment · Reviewer_S7BU · 2025-11-23
> > **Follow up**
> >
> > Thank you for the detailed responses, which address several of my earlier concerns. I have one follow-up question.
> >
> > I agree that modeling the critic distribution using flow matching or diffusion can better capture the multi-modality of value estimates. However, my original question was specifically about whether the mean scalar value learned by DFC is actually more accurate than directly regressing to the expected return. Figure 5 shows that DFC provides a better fit to the full return distribution compared with IQN and C51. But since the policy update only relies on the mean value, is this mean estimate demonstrably more accurate than those produced by IQN, C51, or other non-distributional critic?

---

> > > ### Author Response · Authors · 2025-11-25
> > >
> > > We have now compared the performance in mean estimate as per your valuable suggestion. We would be most grateful if you could kindly let us know whether this addresses your concern.

---

> ### Author Response · Authors · 2025-11-24
> **Re: Follow up**
>
> Thank you for your response and apologies for our previous misunderstanding. We have compared the absolute error between the  mean of our distributional model and the ground-truth mean as follows  including results for the vanilla FQL baseline that directly regresses to the mean. To facilitate comparison across tasks with varying reward scales, the results below are reported on rewards normalized to the range $[0, 1]$
>
> | IQN | FQL |DFC|
> |---|---|---|
> |0.1684|0.1085|0.0432|
>
> The results show that DFC achieves a significant performance advantage, with its normalized mean absolute error being less than half that of the strongest baseline competitor.

---

> ### Comment · Reviewer_S7BU · 2025-11-25
> **Thanks for the detailed rebuttal**
>
> I appreciate the efforts made by the authors. The main concern regarding whether the mean value estimation can be improved by flow-matching critic modeling has been well resolved. Thus, I am happy to raise my score to a 6.

---

### Official Review · Reviewer_SMRe · 2025-10-28

**Soundness:** 2
**Presentation:** 2
**Contribution:** 2
**Rating:** 4
**Confidence:** 2

**Summary:**

This paper addresses the limitation where expressive flow-based policies are often hindered by simple critics that only estimate a single expected return value. To overcome this, this paper introduces the Distributional Flow Critic, which employs flow matching within a two-stage distillation architecture to learn the complete distribution of state-action returns, providing a richer learning signal. Experiments show that integrating DFC achieves strong performance, especially on tasks needing multimodal actions, in both offline and offline-to-online settings.

**Strengths:**

1. The paper's motivation is clearly articulated. It effectively highlights the mismatch between expressive actors and simplistic scalar critics in existing methods, particularly relevant given that flow policies can capture multimodal behavioral distributions.
2. The paper presents convincing results across multiple benchmarks, demonstrating that DFC achieves significant performance improvements over state-of-the-art baselines. The ablation studies are well-designed and validate the necessity of combining Flow Matching and Quantile Regression.
3. The paper is well-presented, making the methodology and results relatively easy to read and understand.

**Weaknesses:**

1. The novelty of this paper is somewhat limited. The main contribution appears to be the integration of distributional RL concepts into the existing Flow Q-Learning framework. Specifically, it involves modeling the return as a distribution to improve the critic's expressiveness and includes architectural modifications to enable the flow policy to leverage these distributional returns.
2. This paper does not provide a theoretical justification or convergence analysis of the proposed two-stage critic. There is no discussion of whether the distillation preserves Bellman consistency or introduces bias in the return distribution.
3. This paper lacks a discussion of computational overhead. While it claims to avoid the BPTT problem, the proposed framework is quite complex, notably requiring the training of two actors and two critics. This raises concerns for the reviewer regarding the potential introduction of significant additional computational load. An analysis of the computational cost would substantially strengthen the paper's claim regarding efficiency.

**Questions:**

1. Can the authors clarify what novel challenges, distinct from standard Distributional RL, arise when introducing distributional returns to flow policies, and how does the proposed architecture address them?
2.The reviewer notes that during the actor update, the expectation of the distributional return provided by the critic is used in the objective. Although this is a common practice in distributional RL, does utilizing only the mean potentially undermine the rich expressiveness of the distributional return itself within the context of a flow policy? For instance, might properties like multimodality in the return distribution be lost or significantly attenuated when reduced to a single expected value for the policy gradient calculation?

---

> ### Author Response · Authors · 2025-11-22
> **Weakness 1-3 and Question 1**
>
> ## W1 & Q1: The novelty of this paper is somewhat limited.
>
> We respectfully clarify that integrating distributional RL into flow-based methods is non-trivial and presents unique algorithmic hurdles distinct from standard DRL. A naive combination fails due to computational instability. Our novelty lies in identifying and solving these specific challenges:
>
> **1. The Specific Challenge: The "BPTT Dilemma"**
> Unlike standard DRL where the Q-network is a simple feed-forward network, a Flow-based Critic defines values via an ODE trajectory.
> * The Problem: Naively applying DRL requires backpropagating gradients through the ODE solver (Backpropagation Through Time, BPTT) to update the actor. This is computationally prohibitive and notoriously unstable, as confirmed by our ablation study on the "Flow-only Critic" (Table 5).
> * Why it is novel: This instability is unique to flow/diffusion policies and does not exist in standard DRL architectures (like C51/IQN), requiring a new architectural solution.
>
> **2. The Solution: Two-Stage Distillation Architecture**
> To resolve this, we designed a Two-Stage Distillation Architecture (Target Flow $\to$ Distilled Quantile Main Critic).
> * Decoupling: We decouple representation learning from the gradient update. The Target Flow Critic learns the complex, multimodal return distribution via flow matching, while the Main Critic distills this knowledge into a quantile representation.
> * Result: This allows the actor to receive stable gradients from the quantile network, bypassing the BPTT instability while retaining the expressivity of the flow model.
>
> **3. Theoretical Depth: Ensuring Operator Contraction**
> The reviewer mentions FQL (Policy Distillation), but **Critic Distillation presents a fundamentally harder theoretical challenge**:
> * Policy Distillation (FQL) is a supervised regression problem (student mimics teacher) and does not affect the convergence of the Bellman operator.
> * Critic Distillation (DFC) inserts a projection step inside the recursive Bellman loop. This is risky because the combined operator—coupling the distributional Bellman operator ($\mathcal{T}^\pi$) with the distillation projection ($\Pi_{W_1}$)—must remain a contraction mapping.
> * Our Contribution: We explicitly prove in the newly added **Proposition 1** that our specific design preserves this contraction property, ensuring stable value iteration. This theoretical guarantee distinguishes our work from simple heuristic integrations.
>
>
> ## W2: "This paper does not provide a theoretical justification or convergence analysis of the proposed two-stage critic. ".
>
> We agree with the reviewer that theoretical grounding is essential. In our revised manuscript, we have added **Proposition 1** to explicitly address the convergence and consistency of our two-stage critic.
>
> * **Contraction Mapping (Proposition 1):** The core concern is whether the distillation step breaks the convergence properties of the Bellman operator. We prove that our distillation process—specifically, projecting the target distribution onto the quantile locations—preserves the contraction property.
>     * Formaly, let $\mathcal{T}^\pi$ be the distributional Bellman operator and $\Pi_{W_1}$ be the projection operator associated with our Quantile Regression distillation. We show that the combined operator $\Pi_{W_1} \mathcal{T}^{\pi}$ is a **contraction mapping** under the Wasserstein metric.
>     * This guarantees that our critic iteration converges to a unique fixed point, providing the theoretical justification the reviewer requested.
>
> * **Addressing Bias:** Regarding the concern about introduced bias:
>     * The distillation is performed via **Quantile Regression**, which is a consistent estimator of the true distribution's quantiles.
>     * By setting $\hat{\tau}$ to the quantile midpoints, we ensure the estimation remains unbiased[3]. Indeed, we discussed the unbiased estimation at the end of the  background section.
>
> The "distillation" here is not an approximation that degrades quality, but a **projection** that retains the essential distributional information required for the actor update while discarding the computational graph needed for ODE differentiation.
>
>
>
> ## W3: "This paper lacks a discussion of computational overhead".
>
> We present the time consumption comparison in Figure 4 in the Appendix B. DFC is only marginally slower than FQL in terms of training time ($1.69$ms  vs $1.42$ms per step), and matches or exceeds the inference efficiency of most baseline methods such as IQL, FQL, FBRAC, and IFQL, with an inference time of $0.63$.

---

> ### Author Response · Authors · 2025-11-22
> **Question 2**
>
> ## Q2: "The reviewer notes that during the actor update ...".
>
>
> We thank the reviewer for this insightful question regarding the utilization of the distributional return.
>
> It is true that we condense the distribution into an expectation for the policy gradient step. However, the "rich expressiveness" of the distribution is not lost; rather, it is the fundamental reason we achieve a stable and accurate learning signal. We justify this design choice with two key points:
>
> **1. Distributional Learning Yields a More Accurate Mean**
> Directly regressing a scalar value to a complex, multimodal target is notoriously difficult and unstable (often leading to "chattering" or convergence to local optima). By explicitly modeling the full distribution, the network captures the underlying modes of the return landscape. This forces the critic to "understand" the complexity of the environment, resulting in a significantly more accurate estimate of the expectation as a byproduct.
>
> To empirically validate this, we measured the **Absolute Error** of the estimated mean against the ground-truth mean (on normalized rewards). As shown in the table below, our method (**DFC**) drastically reduces estimation error compared to the scalar baseline (**FQL**) and classic distributional method (**IQN**):
>
> | Method | IQN | FQL (Scalar Baseline) | **DFC (Ours)** |
> | :--- | :--- | :--- | :--- |
> | **Absolute Error** | 0.1684 | 0.1085 | **0.0432** |
>
> This demonstrates that the "expressiveness" of the distribution is essential for **deriving the correct mean**, even if the policy ultimately maximizes that mean.
>
> **Intuition: Why is the scalar baseline inaccurate?** Consider a "Toy Example" of a robot facing an obstacle: it must go Left or Right (both high returns). A scalar critic averaging these modes might output a value corresponding to going Straight (crash/low return), or oscillate between the two values (the "chattering" phenomenon).  By modeling the multimodal distribution (as shown in Figure 5 appendix B), DFC prevents this aliasing. The distribution acts as a robust auxiliary task, ensuring that when we do calculate the expectation, it is derived from a structurally correct representation, avoiding the instability of scalar regression.
>
> **2. Standard vs. Risk-Sensitive RL Objectives**
> Our current setting follows the standard Reinforcement Learning objective: maximizing the expected cumulative return ($\max \mathbb{E}[R]$). In this context, using the mean is mathematically consistent with the objective.
> While the distributional critic *enables* risk-sensitive applications (e.g., optimizing for CVaR or specific quantiles), our focus in this work is to show that distributional modeling improves performance even on standard benchmarks. The distribution here serves as a **robust representation learner**, stabilizing the gradients provided to the actor.
>
> [1] Gabriel Barth-Maron, Matthew W. Hoffman, David Budden, Will Dabney, Dan Horgan, Dhruva TB, Alistair Muldal, Nicolas Heess, Timothy Lillicrap. Distributed Distributional Deterministic Policy Gradients. In ICLR, 2018.
>
> [2]  Marc G. Bellemare, Will Dabney, and R´emi Munos. A distributional perspective on reinforcement learning. In ICML, 2017.
>
> [3]  Distributional reinforcement learning with quantile regression. AAAI, 2018.

---

> ### Author Response · Authors · 2025-11-26
> **Reminder**
>
> We sincerely thank you for your thoughtful and constructive feedback. In our rebuttal, we have provided detailed responses to your major concerns:
>
> - Statement of our innovation.
> - Theoretical justification of convergence.
> - Comparison of training/inference time consumption across methods.
> - The necessity of the "distributional" design.
>
> We would be most grateful if you could take a moment to review our response. We hope these clarifications sufficiently address your concerns, and we remain available to answer any further questions you may have.

---

### Official Review · Reviewer_oECb · 2025-10-29

**Soundness:** 3
**Presentation:** 3
**Contribution:** 2
**Rating:** 4
**Confidence:** 3

**Summary:**

The paper introduces Distributional Flow Critic (DFC), a critic architecture designed to learn the full distribution of state-action returns rather than a single scalar estimate. DFC extends Flow Q-Learning (FQL) by employing (1) a flow-matching critic that models the return distribution and (2) a quantile regression–based distillation step to stabilize learning. The distilled critic is then used by a flow-policy to maximize expected Q-values. The motivation is that a distributional critic can better match the expressivity of flow-based actors and provide richer learning signals.

**Strengths:**

1. Strong empirical results: DFC achieves impressive performance on challenging benchmarks such as antmaze-giant-singletask and puzzle-4x4-singletask, outperforming strong diffusion and flow-based baselines.
2. Stable architecture: The proposed two-stage critic design successfully addresses the instability of directly training a flow-based critic by leveraging distributional distillation.
3. Comprehensive evaluation: Experiments cover both offline and offline-to-online reinforcement learning settings, demonstrating robustness and sample efficiency.

**Weaknesses:**

1. Limited conceptual novelty: The distillation of a critic distribution from a flow-based model is conceptually similar to existing policy distillation techniques (e.g., FQL). The contribution mainly lies in adapting known ideas to the critic side.
2. Dependency on FQL performance: DFC is tightly coupled with FQL; it improves upon FQL where FQL performs well but fails in the same settings where FQL struggles. This raises questions about whether the critic is truly the bottleneck.
3. Lack of ablation clarity: While the ablation in Table 5 compares “Flow-only” and “Distributional-only” critics, it would be stronger if supported by explicit variance metrics or training curves demonstrating instability versus convergence.
4. Insufficient empirical evidence for bottleneck claim: The claim that “the critic is the bottleneck” is asserted but not convincingly demonstrated through diagnostics such as critic loss analysis or policy entropy trends.

**Questions:**

1. Figure 2 visualization: The figure effectively summarizes improvements but is visually overwhelming. The authors could highlight only a subset of key representative tasks to enhance readability.
2. Derivation of sample $z_j^{'}$ is slightly confusing since in Equation 4 it is written as function evaluation while in the text it is mentioned to be computed by solving the ODE
3. Is this exclusive to flow policy? The central claim—“a distributional critic unlocks superior performance for flow-based policies”—is convincing but overly restrictive. The authors should clarify whether DFC can also serve as a general-purpose critic for non-flow actors (e.g., ReBRAC, IQL). Testing on at least one non-flow baseline would strengthen generality.
4. Many prior works stabilize critic learning through ensembles. It would be helpful to specify whether DFC uses any ensemble mechanism, or whether the combination of flow matching and quantile regression alone ensures stability. Empirical comparison with ensemble-based baselines would clarify this.
5. The meaning of the arrows (“→”) is unclear and should be explained—do they indicate training progression or pre/post fine-tuning results? Additionally, the claim that “FC suffers from training instability” should be supported with quantitative evidence such as standard deviation across seeds or critic loss variance
6. The paper attributes FQL’s failure cases to the critic bottleneck. However, some tasks (e.g., large antmaze variants) remain unsolved even with distributional critics. A more nuanced discussion—e.g., whether performance is limited by actor expressivity, data coverage, or Bellman extrapolation—would be appreciated.
7. The claim in the paper is that the distributional Bellman operator will help the performance. Could you also combine other existing distributional methods with the FQL for a fair comparison to prove that DFC is novel?

---

> ### Author Response · Authors · 2025-11-22
> **Weakness 1-3**
>
> We thank the reviewer for the constructive comments.
> ## W1:  Limited conceptual novelty.
>
> We respectfully disagree with the assessment that the novelty is limited to merely "adapting known ideas." We argue that applying distillation to the critic in a Distributional RL context presents a unique theoretical challenge that does not exist in policy distillation.
>
> **Conceptual Novelty: The Challenge of Operator Contraction**
> * **Policy Distillation (e.g., FQL)** is typically a straightforward supervised regression problem where the student mimics the teacher's output. It does not inherently interact with the recursive stability of a Bellman operator.
> * **Critic Distillation (DFC)**, however, inserts a projection step *inside* the recursive Bellman update loop. This is non-trivial because the distillation process must be carefully designed to ensure that the combined operator—coupling the **distributional Bellman operator** ($\mathcal{T}^\pi$) with the **distillation projection** ($\Pi_{W_1}$, w.r.t. the quantile loss)—remains a **contraction mapping**. If this property is violated, the value iteration will diverge.
>
> **The Innovation: Proposition 1 (Theoretical Guarantee)**
> In our new pdf, we have added **Proposition 1** to formally address this. We prove that our specific design—using Quantile Regression to distill the target flow distribution—preserves the contraction property under the Wasserstein metric.
> * This theoretical guarantee validates that DFC is not just an engineering adaptation, but a mathematically sound framework for Generative Distributional RL.
> * Standard policy distillation techniques do not address (and do not require) such fixed-point convergence proofs involving Bellman operators.
>
>
> ## W2 & Q6: Dependency on FQL performance.
>
> The reviewer correctly observes that DFC is coupled with FQL and that its performance is contingent on the underlying policy's capability. We would like to clarify the scope and role of our contribution:
>
> **1. DFC is a Critic Architecture, Not a Standalone Algorithm.**
> DFC is proposed as a novel **Critic Architecture** designed to provide a richer, high-fidelity gradient signal. It is not intended to replace the actor mechanism itself. Therefore, it naturally relies on the base actor (in this case, FQL) to be at least fundamentally viable for the given task.
>
> **2. Differentiating Failure Modes vs. Bottlenecks.**
> * **When FQL fails (Exploration Issue):** In tasks where the baseline FQL fails completely (e.g., due to lack of exploration or inability to stitch trajectories in specific OGBench tasks), DFC cannot magically recover performance. This is expected behavior: a better critic cannot guide a policy that cannot find *any* valid path.
> * **When FQL works (Critic Bottleneck):** However, in the majority of tasks where the FQL actor *is* viable, DFC consistently boosts performance (often by $>10\%$) and improves stability. This validates our core hypothesis: when the actor has the capacity to learn, the **scalar critic was indeed the bottleneck**. By removing this bottleneck with our distributional design, we unleash the full potential of the flow-based policy.
>
> **3. Generalizability.**
> While we implemented DFC on top of FQL to solve the specific BPTT stability challenge in flow-based RL, the principle of **"Generative Critic for Generative Policy"** is general. We believe this architecture could be adapted to enhance other generative policy methods (e.g., Diffusion-QL) where similar expressivity gaps exist.
>
> ## W3: Lack of ablation clarity.
>
> **Variance Analysis:** We quantitatively analyzed stability using the results from Table 5 Aggregating across the entire benchmark suite, the Flow-only Critic (FC) exhibits an average standard deviation of $7.45$ whereas DFC significantly reduces this to $4.07$, demonstrating a $1.8+$ times improvement in stability globally.

---

> ### Author Response · Authors · 2025-11-22
> **Weakness 4 and Question 1-7**
>
> ## W4: Insufficient empirical evidence for bottleneck claim.
>
> The reviewer asks for a demonstration that the critic is the bottleneck. We believe the strongest evidence lies in the **Performance Gap** demonstrated in our Ablation Study:
> * **Multimodality":** As argued in standard Distributional RL literature, a scalar critic inevitably averages out multimodal returns. We demonstrate  the ground truth distribution and  learned return distributions of C51, IQN, and DFC in Figure 5 in our new pdf. The results verify that the return landscape in some scenario is indeed **multimodal**. Crucially, DFC achieves a **significantly superior fit** (quantitively over 2.5 times measured by the lowest 1-Wasserstein distance) compared to C51 and IQN.
> * **The "Expressivity Gap":** in our **Ablation Study (Table 5)**. The "Distributional-only Critic" (stable but limited expressivity) and "Flow-only Critic" (expressive but unstable) both significantly underperform our full DFC model. This confirms that the synergy of distributional modeling and flow-based expressivity is a performance prerequisite, not just a stylistic choice.
> * **Empirical Proof:** Our ablation in **Table 5**  (appendix B) serves as the diagnostic:
>     1.  Using a stable but limited critic (**Distributional-only/DC**) yields suboptimal results, often performing similarly to the baseline.
>     2.  Using a standard scalar critic (**FQL**) yields lower performance.
>     3.  Only when we introduce the **Distributional Flow Critic (DFC)**—which uniquely combines high expressivity with training stability—do we see significant state-of-the-art gains.
>     The observation that increasing the critic's representational capacity (while strictly maintaining stability via our distillation operator) directly leads to substantial performance gains provides strong empirical proof that the critic's expressivity was indeed the limiting factor (the bottleneck).
>
>
>
> ## Q1: Figure 2 visualization.
>
> We have highlighted the tasks by subsets and refurnished the figure, hoping to erase the problem. Thanks for advising.
>
> ## Q2: Derivation of sample $z_j'$.
>
> $Z_{\tilde\phi}$ is a flow-matching critic, which theoretically indicates solution of an ODE function. $z_j'$ is the output of critic network $Z_{\tilde\phi}$ from the previous iteration.
>
> ## Q3: Is this exclusive to flow policy?
>
> We present ablation experiments in Table 8 in our revised manuscript, which evaluate our model's performance across different policy architectures. The results show that, our method consistently outperforms the vanilla critic baseline on both locomotion and manipulation tasks on OGBench benchmarks. This demonstrates the robustness of our approach to variations in the policy model and underscores the advantage of using a distributional critic in diverse environments.
>
> ## Q4: Stability and comparison with ensembled baselines.
>
> We conducted experiments comparing our approach to ensembled critic baselines, evaluating both 2-ensemble and 4-ensemble Q-networks of FQL (see Table 7 in our revised manuscript). Across diverse environment, including D4RL's adroit tasks and OGBench state-based benchmarks, our DFC method consistently delivers the strongest performance, outperforming ensemble-based critics in nearly all cases. These results demonstrate that DFC achieves clear advantages over traditional ensemble methods, confirming the effectiveness and robustness of our distributional critic design.
>
> We show that the combination of the distillation projection induced by quantile regression with the Bellman operator is a contraction, guaranteeing convergence. For further details see **proposition 1**.
>
> ## Q5: "The meaning of the arrows (“→”) and questions on FC...  standard deviation across seeds or critic loss variance".
>
> The meaning of $\to$ illustrates the performance improvement during the offline-to-online scheme. The performance to the left of $\to$ is the result under offline setting while that to the right is the result after 100k epochs of online fine-tuning. We have added the description to the captions. For instability of FC, it is empirically shown in Table 5 with extensive tasks. FC exhibits an average standard deviation of $7.45$ whereas DFC significantly reduces this to $4.07$, demonstrating a $1.8+$ times improvement in stability globally.
>
>
>
> ## Q7:  Combination of the other existing distributional methods.
>
> We have provided a comprehensive evaluation of DFC against established distributional approaches, C51 and IQN, as summarized in Table 6 in our revised manuscript. Using identical actor architectures across all methods, DFC consistently surpasses both C51 and IQN on D4RL adroit tasks and OGBench state-based benchmarks, achieving an average improvement of approximately $14\%$. These results highlight the effectiveness and generality of our approach.

---

> > ### Author Response · Authors · 2025-11-23
> > **Table 8**
> >
> > Table 8. **Ablation study of critic structure.**
> > We plugin the DF critic to ReBRAC and IQL. The results show that, our method consistently outperforms the vanilla critic baseline on both locomotion and manipulation tasks on OGBench benchmarks.
> >
> > | env_name               | ReBRAC | ReBRAC + DF Critic | IQL  |IQL + DF Critic  |
> > |------------------------|--------|--------------------|------|------|
> > | antmaze-large-navigate-singletask-v0       |  $91\pm10$ |  $91\pm1$ |$48\pm9$|$67\pm4$|
> > | antmaze-giant-navigate-singletask-v0       |  $27\pm22$ |  $74\pm5$ |$0\pm0$|$0\pm0$|
> > | humanoidmaze-medium-navigate-singletask-v0 |  $16\pm9$ | $23\pm1$  |$32\pm7$|$43\pm3$|
> > | humanoidmaze-large-navigate-singletask-v0  |  $2\pm1$ |  $5\pm10$ |$3\pm1$|$5\pm1$|
> > | antsoccer-arena-navigate-singletask-v0     |  $0\pm0$ |  $23\pm10$ |$3\pm2$|$14\pm1$|
> > | cube-single-play-singletask-v0             |  $92\pm4$ |  $100\pm0$ |$85\pm8$|$100\pm0$|
> > | cube-double-play-singletask-v0             |  $7\pm3$ | $8\pm0$  |$1\pm1$|$4\pm1$|
> > | scene-play-singletask-v0                   |  $50\pm13$ | $99\pm1$  |$12\pm3$|$64\pm10$|
> > | puzzle-3x3-play-singletask-v0              |  $2\pm1$ |  $98\pm0$ |$2\pm1$|$4\pm1$|
> > | puzzle-4x4-play-singletask-v0              |  $10\pm3$ |  $3\pm10$ |$5\pm2$|$11\pm2$|

---

> ### Author Response · Authors · 2025-11-23
> **Table 6-7**
>
> Table 6: **Ablation study of distributional methods.**
> We compare the DFC model against other distributional methods: C51 and IQN. We all use the same flow-actor structure according to FQL.
> | env_name  | C51          | IQN          | DFC          |
> |---|---|---|-----|
> | pen-human-v1  | $\mathbf{69} \pm 8$   | $\mathbf{69} \pm 3$   | $61 \pm 7$   |
> | pen-cloned-v1   | $67 \pm 9$   | $80 \pm 11$  | $\mathbf{90} \pm 7$   |
> | pen-expert-v1    | $110 \pm 3$  | $118 \pm 19$ | $\mathbf{147} \pm 1$  |
> | door-human-v1          | $0 \pm 0$    | $0 \pm 0$    | $\mathbf{6} \pm 3$    |
> | door-cloned-v1         | $0 \pm 0$    | $0 \pm 0$    | $\mathbf{6} \pm 5$    |
> | door-expert-v1         | $104 \pm 1$  | $\mathbf{105} \pm 0$  | $\mathbf{105} \pm 0$  |
> | hammer-human-v1        | $3 \pm 1$    | $2 \pm 1$    | $\mathbf{4} \pm 1$    |
> | hammer-cloned-v1       | $0 \pm 0$    | $0 \pm 0$    | $\mathbf{14} \pm 13$  |
> | hammer-expert-v1       | $122 \pm 1$  | $121 \pm 7$  | $\mathbf{128} \pm 0$  |
> | relocate-human-v1      | $0 \pm 0$    | $0 \pm 0$    | $0 \pm 0$    |
> | relocate-cloned-v1     | $0 \pm 0$    | $0 \pm 0$    | $\mathbf{1} \pm 0$    |
> | relocate-expert-v1     | $103 \pm 0$  | $103 \pm 0$  | $\mathbf{109} \pm 0$  |
> | antmaze-large-navigate-singletask-v0       | $1 \pm 1$    | $30 \pm 8$  | $\mathbf{90} \pm 1$   |
> | antmaze-giant-navigate-singletask-v0       | $0 \pm 0$    | $0 \pm 0$    | $\mathbf{25} \pm 12$  |
> | humanoidmaze-medium-navigate-singletask-v0 | $2 \pm 4$    | $33 \pm 4$   | $\mathbf{35} \pm 9$   |
> | humanoidmaze-large-navigate-singletask-v0  | $0 \pm 0$    | $8 \pm 6$   | $\mathbf{14} \pm 6$   |
> | antsoccer-arena-navigate-singletask-v0     | $2 \pm 2$    | $27 \pm 6$  | $\mathbf{52} \pm 7$   |
> | cube-single-play-singletask-v0        | $4 \pm 2$    | $98 \pm 6$   | $\mathbf{100} \pm 0$  |
> | cube-double-play-singletask-v0        | $0 \pm 0$    | $24 \pm 9$   | $\mathbf{49} \pm 2$   |
> | scene-play-singletask-v0         | $0 \pm 0$    | $1 \pm 0$    | $\mathbf{96} \pm 3$   |
> | puzzle-3x3-play-singletask-v0       | $1 \pm 4$    | $0 \pm 0$    | $\mathbf{21} \pm 5$   |
> | puzzle-4x4-play-singletask-v0         | $0 \pm 0$    | $\mathbf{23} \pm 2$   | $20 \pm 3$   |
>
>
> Table 7: **Ablation study of ensemble methods.**
> We compare the DFC model against FQL with ensembled value models. DFC performs better on all the tasks listed.
> | env_name               | DFC          | FQL (\# $Q=2$ )         | FQL (\# $Q=4$)         |
> |------------------------|--------------|----------------------------|----------------------------|
> | pen-human-v1           | $\mathbf{61} \pm 7$   | $53 \pm 6$         | $39 \pm 12$                |
> | pen-cloned-v1          | $\mathbf{90} \pm 7$   | $74 \pm 11$        | $76 \pm 15$                |
> | pen-expert-v1          | $\mathbf{147} \pm 1$  | $142 \pm 6$        | $145 \pm 2$                |
> | door-human-v1          | $\mathbf{6} \pm 3$    | $6 \pm 3$          | $0 \pm 0$                  |
> | door-cloned-v1         | $\mathbf{6} \pm 5$    | $2 \pm 1$          | $1 \pm 1$                  |
> | door-expert-v1         | $\mathbf{105} \pm 0$  | $104 \pm 1$        | $105 \pm 0$                |
> | hammer-human-v1        | $\mathbf{4} \pm 1$    | $3 \pm 1$                  | $1 \pm 1$                  |
> | hammer-cloned-v1       | $\mathbf{14} \pm 13$  | $11 \pm 9$          | $4 \pm 1$                  |
> | hammer-expert-v1       | $\mathbf{128} \pm 0$  | $125 \pm 3$          | $127 \pm 1$                |
> | relocate-human-v1      | $\mathbf{0} \pm 0$    | $0 \pm 0$            | $0 \pm 0$                  |
> | relocate-cloned-v1     | $\mathbf{1} \pm 0$    | $-0 \pm 0$           | $-0 \pm 0$                 |
> | relocate-expert-v1     | $\mathbf{109} \pm 0$  | $107 \pm 1$          | $109 \pm 1$                |
> | antmaze-large-navigate-singletask-v0       | $\mathbf{90} \pm 1$   | $80 \pm 8$      | $56 \pm 12$                |
> | antmaze-giant-navigate-singletask-v0       | $\mathbf{25} \pm 12$  | $4 \pm 5$       | $17 \pm 4$                 |
> | humanoidmaze-medium-navigate-singletask-v0 | $\mathbf{35} \pm 9$   | $19 \pm 12$     | $22 \pm 14$                |
> | humanoidmaze-large-navigate-singletask-v0  | $\mathbf{14} \pm 6$   | $7 \pm 6$           | $6 \pm 5$                  |
> | antsoccer-arena-navigate-singletask-v0     | $\mathbf{52} \pm 7$   | $39 \pm 6$    | $47 \pm 6$                 |
> | cube-single-play-singletask-v0             | $\mathbf{100} \pm 0$  | $97 \pm 2$    | $98 \pm 2$                 |
> | cube-double-play-singletask-v0             | $\mathbf{49} \pm 2$   | $36 \pm 6$    | $56 \pm 12$                |
> | scene-play-singletask-v0                   | $\mathbf{96} \pm 3$   | $76 \pm 9$    | $83 \pm 4$                 |
> | puzzle-3x3-play-singletask-v0              | $\mathbf{21} \pm 5$   | $16 \pm 5$    | $19 \pm 5$           |
> | puzzle-4x4-play-singletask-v0              | $\mathbf{20} \pm 3$   | $11 \pm 3$            | $6 \pm 3$             |

---

> ### Author Response · Authors · 2025-11-26
> **Reminder**
>
> We sincerely thank you for your thoughtful and constructive feedback. In our rebuttal, we have provided detailed responses to your major concerns:
>
> - Statement of our innovation.
> - Clarification of the scope and role of our contribution.
> - Variance analysis.
> - The demonstration of the performance gap.
> - Figure 2 visualization.
> - Derivation of sample $z_j'$.
> - Theoretical justification of convergence and comparison with ensembled baselines.
> - The meaning of "$\to$".
> - Combination of the other existing distributional methods.
>
> We would be most grateful if you could take a moment to review our response. We hope these clarifications sufficiently address your concerns, and we remain available to answer any further questions you may have.

---

### Official Review · Reviewer_N8K2 · 2025-11-03

**Soundness:** 2
**Presentation:** 2
**Contribution:** 2
**Rating:** 2
**Confidence:** 2

**Summary:**

This paper proposes Distributional Flow Critic (DFC), a novel critic architecture that models the full return distribution using flow matching for offline and offline-to-online reinforcement learning. DFC addresses the information bottleneck in flow-based policies caused by scalar critics. The method distills a multi-step target flow critic into a single-step critic via quantile regression, avoiding backpropagation through ODE solvers. Core contributions include: (1) A two-stage critic design mitigating training instability (§3.2, Fig. 1), (2) Integration with Flow Q-Learning actors (§3.3), and (3) Empirical gains over baselines on 73 D4RL/OGBench tasks, with 10% average improvement in offline RL and superior fine-tuning performance.

**Strengths:**

- Novel Distributional Critic Architecture. The two-stage flow critic elegantly circumvents backpropagation through ODEs while capturing full return distributions.
- Rigorous Multi-Benchmark Validation. Extensive tests on 73 tasks show consistent gains over FQL on OGBench manipulation.
- Effective Offline-to-Online Transfer. Fine-tuning with online data improves FQL effectively on long-horizon tasks.

**Weaknesses:**

- lack of an ablation study to illustrate the effectivenesss of different components. For example, how do different expectile hyperparameter $\hat\tau$ and sample numbers $M$ influence the performance?
- Modeling the critic $Q(s,a)$ as a distribution is quite eccentric. That is, the definition of $Q(s,a)$ is the expectation of the returned value, a number instead of a distribution.  This manner can also lead to unstable training due to the much more severe error propagation of the TD estimator, as shown in  Eq.4. If the author wants to claim the necessity of such a critic-modeling method, an intuitive toy example/experiments should be performed.
- The time-consuming comparison between the proposed DFC and other flow policy-based methods should be presented.

**Questions:**

- Can you theoretically show how the critic trained in DFC will converge?

---

> ### Author Response · Authors · 2025-11-22
> **Weakness 1 and 3**
>
> ## W1 : "lack of an ablation study to illustrate the effectivenesss of different components. "
>
> We thank the reviewer for emphasizing the importance of validating our design choices. We address this concern in two parts: first, by highlighting the **architectural ablation already included** in our submission (demonstrating the necessity of our two-stage design), and second, by providing new ablations regarding the distributional settings.
>
> **1. Effectiveness of Architectural Components (FC & DC)**
> To illustrate the effectiveness of the specific components in our method, we refer the reviewer to **Section 5.3 and Table 5 (Appendix B)** of our paper. Here, we explicitly compared our full model against two simplified variants to validate our core hypothesis:
>
> Necessity of Distillation (vs. Flow-only Critic): We compared against a Flow-only Critic (FC), which attempts to learn without the separate distillation step with a distributional critic. As hypothesized, this variant suffers from significant training instability and high variance ($7.45$ vs $4.07$). This confirms that our main critic is essential to avoid the instability of backpropagating through the ODE solver (BPTT).
>
> Necessity of Flow Matching (vs. Distributional-only Critic): We compared against a Distributional-only Critic (DC), which uses a standard Quantile Regression critic without the flow-based target generation. While training is stable, it yields suboptimal performance. This confirms that the expressivity of the flow-based target is critical for modeling complex return distributions in RL, which traditional distributional critics fail to capture.
>
> Conclusion: Our full DFC model synergizes these components to achieve state-of-the-art performance, effectively solving the dilemma between expressivity and stability.
>
> **2. Ablation on Distributional Hyperparameters on $\hat{\tau}$ and sample size $M$.**
>
> We appreciate the reviewer's suggestion to provide a more detailed analysis of these components. We have added **Table 9** to the appendix to present these ablation results.
>
> **i. Clarification on the Quantile Hyperparameter $\hat{\tau}$.**
> We would like to clarify that in our framework, $\hat{\tau}$ is **not** a manually tunable hyperparameter. Following the standard Quantile Regression formulation [1,3], $\hat{\tau}$ represents the *quantile midpoints* required to ensure unbiased estimation of the distribution. It is deterministically derived from the number of atoms $M$ as:
> $$\hat{\tau}_i = \frac{2i-1}{2M}, \quad \text{for } i \in \{1, \dots, M\}$$
> Therefore, $\hat{\tau}$ is inextricably linked to $M$, and ablating it independently violates the theoretical consistency of the quantile regression loss.
>
> **ii. Ablation on the Number of Quantiles $M$.**
> To address the reviewer's comment on sample size, we conducted additional experiments comparing our default setting ($M=51$) against a finer-grained resolution ($M=101$). The results are summarized in the newly added **Table 9**.
>
> Our empirical findings indicate that the method is **robust** to the choice of $M$:
> * **Performance Saturation:** While increasing $M$ theoretically allows for a more precise approximation of the return distribution, empirically, we observed that $M=51$ is sufficient to capture the necessary distributional information for effective policy learning.
> * **Marginal Differences:** The performance differences between $M=51$ and $M=101$ were statistically marginal and varied across environments without a consistent winner.
>
> Consequently, we adopted $M=51$ as a deliberate design choice to strike an optimal balance between **distributional expressivity** and **computational efficiency**.
>
> **3. Additional Ablations (Appendix B)**: We have expanded our experimental analysis to isolate the effectiveness of our design. (1) vs. Distributional Baselines (Table 6): DFC outperforms established methods (C51, IQN) by ~14% on average using identical actors. (2) vs. Ensemble Critics (Table 7): DFC consistently surpasses 2-ensemble and 4-ensemble baselines, proving the gain stems from distributional modeling rather than parameter count. (3) Visualization (Figure 5): DFC achieves 2x lower 1-Wasserstein distance than baselines, accurately capturing multimodal returns. (4) Generality & Efficiency: We demonstrate DFC's robustness across different policy architectures (Table 8).
>
> ## W3: "The time-consuming comparison between the proposed DFC and other flow policy-based methods should be presented."
>
> We present the time consumption comparison in Figure 4 of Appendix B. DFC is only marginally slower than FQL in terms of training time($1.69$ms vs $1.42$ms per step), and matches or exceeds the inference efficiency of most baseline methods such as IQL, FQL, FBRAC, and IFQL, with an inference time of $0.63$.

---

> > ### Author Response · Authors · 2025-11-23
> > **Table 9**
> >
> > Table 9: **Ablation study of sample numbers $M$.** We ablate $M$ in $\{51,101\}$ following [2]. The best scores are highlighted in **bold**.
> >
> > | env_name               | $M=51$          | $M=101$    |
> > |------------------------|--------------|----------------|
> > | pen-human-v1           | $\mathbf{61} \pm 7$   | $84 \pm 3$     |
> > | pen-cloned-v1          | $\mathbf{90} \pm 7$   | $77 \pm 5$     |
> > | pen-expert-v1          | $\mathbf{147} \pm 1$  | $130 \pm 12$   |
> > | door-human-v1          | $\mathbf{6} \pm 3$    | $8 \pm 2$      |
> > | door-cloned-v1         | $\mathbf{6} \pm 5$    | $1 \pm 1$      |
> > | door-expert-v1         | $105 \pm 0$  | $\mathbf{106} \pm 0$    |
> > | hammer-human-v1        | $4 \pm 1$    | $3 \pm 1$      |
> > | hammer-cloned-v1       | $\mathbf{14} \pm 13$  | $1 \pm 1$      |
> > | hammer-expert-v1       | $\mathbf{128} \pm 0$  | $123 \pm 1$    |
> > | relocate-human-v1      | ${0} \pm 0$    |  $0 \pm 0$    |
> > | relocate-cloned-v1     | $\mathbf{1} \pm 0$    |  $0 \pm 0$            |
> > | relocate-expert-v1     | $\mathbf{109} \pm 0$  | $107 \pm 1$        |
> > | antmaze-large-navigate-singletask-v0       | $90 \pm 1$   |   $\mathbf{93}\pm 2$    |
> > | antmaze-giant-navigate-singletask-v0       | $\mathbf{25} \pm 12$  |   $20 \pm 6$   |
> > | humanoidmaze-medium-navigate-singletask-v0 | $\mathbf{35} \pm 9$   |    $33\pm 8$     |
> > | humanoidmaze-large-navigate-singletask-v0  | $14 \pm 6$   |      $14 \pm 5$          |
> > | antsoccer-arena-navigate-singletask-v0     | $\mathbf{52} \pm 7$   |      $50\pm3$          |
> > | cube-single-play-singletask-v0             | $100 \pm 0$  |      $100 \pm 0$          |
> > | cube-double-play-singletask-v0             | $49 \pm 2$   |  $\mathbf{74}\pm 8$   |
> > | scene-play-singletask-v0                   | $96 \pm 3$   | $\mathbf{99} \pm 1$  |
> > | puzzle-3x3-play-singletask-v0              | $21 \pm 5$   |    $\mathbf{52} \pm 3$   |
> > | puzzle-4x4-play-singletask-v0              | $\mathbf{20} \pm 3$   |      $18 \pm 5$   |

---

> ### Author Response · Authors · 2025-11-22
> **Weakness 2 and Question 1**
>
> ## W2: "Modeling the critic Q(s,a) as a distribution is quite eccentric..."
>
> We appreciate the reviewer's scrutiny. We respectfully clarify that modeling the return distribution is neither eccentric nor unstable; rather, it is a prerequisite for accurate value estimation in complex environments.
>
> **1. Established Paradigm (Not "Eccentric")**
> While $Q^\pi$ is defined as an expectation, Distributional RL (DRL)[1,2,3] is a widely established paradigm. It explicitly models the random variable $Z^\pi(s,a)$ because capturing higher-order moments (variance, multimodality) provides a richer learning signal that stabilizes the representation learning, which a scalar mean averages out.
>
> **2. Why Distributional? (Accuracy, Intuition, and "Chattering")**
> The reviewer asks for the necessity of this approach.
>
>  We argue that learning the full distribution yields a significantly **more accurate mean estimate** than directly regressing to the mean. Empirical Evidence (New Experiment): We measured the absolute error between the estimated mean and the ground-truth mean (normalized rewards). DFC reduces the estimation error by ~60% compared to the scalar baseline (FQL):
>
> | IQN | FQL |DFC(ours)|
> |---|---|---|
> |0.1684|0.1085|0.0432|
>
> Intuition & "Chattering": Why is the scalar baseline inaccurate? Consider a "Toy Example" of a robot facing an obstacle: it must go Left or Right (both high returns). A scalar critic averaging these modes might output a value corresponding to going Straight (crash/low return), or oscillate between the two values (the "chattering" phenomenon).
>
> Our Solution: By modeling the multimodal distribution (as shown in Figure 5 appendix B), DFC prevents this aliasing. The distribution acts as a robust auxiliary task, ensuring that when we do calculate the expectation, it is derived from a structurally correct representation, avoiding the instability of scalar regression.
>
>
>
> ## Q1: Can you theoretically show how the critic trained in DFC will converge?
>
> In our revised manuscript, we have added **Proposition 1** to explicitly prove the convergence  of our two-stage critic.
>
> * **Contraction Mapping (Proposition 1):** The core concern is whether the distillation step breaks the convergence properties of the Bellman operator. We prove that our distillation process—specifically, projecting the target distribution onto the quantile locations—preserves the contraction property.
>     * Let $\mathcal{T}^\pi$ be the distributional Bellman operator and $\Pi_{W_1}$ be the projection operator associated with our Quantile Regression distillation. We show that the combined operator $\Pi_{W_1} \mathcal{T}^\pi$ is a **contraction mapping** under the Wasserstein metric.
>     * This guarantees that our critic iteration converges to a unique fixed point, providing the theoretical justification the reviewer requested.
>
>
>
> [1]  Implicit Quantile Networks for Distributional Reinforcement Learning.  ICML, 2018.
>
> [2] A distributional perspective on reinforcement learning. ICML, 2017.
>
> [3]  Distributional reinforcement learning with quantile regression. AAAI, 2018.

---

> ### Author Response · Authors · 2025-11-26
> **Reminder**
>
> We sincerely thank you for your thoughtful and constructive feedback. In our rebuttal, we have provided detailed responses to your major concerns:
>
> - The necessity of our two-stage design, supplemented with new ablation studies on the distributional modeling settings;
> - Comparison of training/inference time consumption across methods;
> - The benefits of explicitly modeling the full return distribution rather than only its expectation;
> - Theoretical justification for the two-stage critic convergence.
>
> We would be most grateful if you could take a moment to review our response. We hope these clarifications sufficiently address your concerns, and we remain available to answer any further questions you may have.

---

### Comment · Area_Chair_y1ar · 2025-11-23
**Subject: Reminder: Update Your Reviews After Rebuttal**

Dear Reviewers,

The authors have submitted their rebuttal, and we now require your follow-up assessments to move the decision process forward. Please review the authors’ responses and update your evaluations accordingly.

Your prompt follow-up is necessary for us to finalize the meta-review.
Kindly submit your updates as soon as possible.

Best,
Area Chair

---

### Author Response · Authors · 2025-12-01
**Summary of Our Rebuttal**

Dear Area Chair,

Thank you for the time and effort you have devoted to evaluating our submission, the reviews, and our rebuttal. To help reduce your workload, we provide a brief summary of the key consensus reached by the reviewers, followed by our response to common concerns and details on how our revised manuscript addresses them.

Regrettably, prior to the system rollback, **only Reviewer S7BU** had provided a response to our rebuttal. S7BU explicitly stated that their main concern was resolved and **raised their score from 4 to 6** as evidenced by the discussion history (**Nov 25**).

### Summary of Strengths

* **Novelty & Stability (Reviewers N8K2, oECb, S7BU)**: Reviewers praised the "elegant" and "stable" two-stage flow critic architecture. They recognized that our design effectively circumvents the instability of direct training and ODE backpropagation.
* **Strong Empirical Performance (All Reviewers)**: There is a consensus on the robust evaluation. Reviewers highlighted the significant and consistent gains over SOTA baselines (e.g., FQL) across challenging benchmarks (OGBench, D4RL, AntMaze).
* **Clear Motivation (Reviewers SMRe, S7BU)**: Reviewers appreciated the well-articulated motivation regarding the mismatch between expressive flow policies and scalar critics, as well as the paper's clarity.

### Response to Common Concerns

1. **Necessity of Distributional Modeling & Validation of the "Critic Bottleneck" (Reviewers N8K2, S7BU, oECb):**
We clarify that direct scalar regression is unstable, causing the scalar critic to become the primary performance bottleneck (addressing oECb). We emphasize that modeling the full distribution acts as a robust auxiliary task that drastically improves mean estimation accuracy. Specifically, DFC reduces the absolute estimation error by >60% compared to the scalar baseline (FQL: 0.1085 $\to$ DFC: 0.0432), proving that the inaccurate scalar critic was indeed the bottleneck.

(**Note**: Reviewer S7BU explicitly acknowledged this resolution and raised their score to 6 prior to the system rollback.)





2. **Theoretical Guarantees and Convergence (Reviewers SMRe, N8K2):**
We clarify that the two-stage distillation process is theoretically sound. We emphasize the addition of **Proposition 1** in the revised manuscript, which explicitly proves that our method preserves Bellman consistency and guarantees convergence, addressing concerns about theoretical justification.

3. **Computational Efficiency (Reviewers SMRe, N8K2):**
We clarify that the perceived complexity does not translate to significant runtime overhead. We emphasize our runtime analysis (**Appendix B, Figure 4**), demonstrating that DFC is only marginally slower in training ($1.69$ms vs $1.42$ms) and matches SOTA baselines in inference ($0.63$ms), ensuring practical scalability.

4. **Comprehensive Ablations & Robustness (Reviewers N8K2, oECb):**

We clarify the source of our performance gains through extensive new ablations in Appendix B & C. We emphasize three key findings that validate our design choices:
* Robustness to Hyperparameters (Table 9): Addressing N8K2, we show performance saturates at $M=51$ (vs. $M=101$), confirming DFC is robust to quantile resolution and balances expressivity with efficiency.
* Modeling vs. Parameters (Table 7): Addressing oECb, DFC consistently outperforms 4-ensemble baselines, proving that gains stem from accurate distributional modeling (2x lower Wasserstein distance, Figure 5) rather than merely increased parameter count.
* Superiority over Baselines (Table 6): DFC surpasses established distributional methods (C51, IQN) by ~14% using identical actors, further validating the architecture's effectiveness.

---

### Meta-Review · Area_Chair_GdJG · 2026-01-06

**Summary:**

The paper proposes DFC, a two-stage distributional critic for flow-based offline/offline-to-online RL (flow-matching target critic distilled into a one-step quantile critic), claiming that scalar critics bottleneck expressive flow actors and that distillation avoids unstable/expensive backprop through ODE solvers; however, the main submission does not establish a clear, rigorous motivation-to-method-to-evidence chain, and the core contribution reads as an over-complicated engineering stack without a sufficiently convincing justification.

**Reviewer Concerns:**

Key concerns remain outstanding for publication quality: the manuscript’s conceptual exposition around “modeling 𝑄(𝑠,𝑎)
as a distribution” is confusing and does not cleanly connect the learned return distribution, the distilled critic, and the actor objective; the central “critic bottleneck” claim is asserted rather than diagnosed with targeted evidence (e.g., critic error/stability analyses that rule out actor/data/extrapolation limits); and stability/compute tradeoffs are not transparently demonstrated in the main paper given the added components and bootstrapped distributional TD.

**Reviewer Scores:**

Even with discussion, I would not expect the panel to converge to acceptance because the rebuttal-style additions do not fix the fundamental publication issues in the main manuscript: low-confidence positive readings may increase slightly with extra tables/propositions, but the strongest negative assessment reflects missing clarity and mechanistic evidence rather than minor gaps, so the overall evaluation should remain below threshold.

---

### Decision · Program_Chairs · 2026-01-26

Reject